# DiscoForcing: A Unified Framework for Real-Time Audio-Driven Character Control with Diffusion Forcing

Kaiyang Ji [1 2 *]   Bingsheng Qian [1 2 *]   Binghuan Wu [1]   Kangyi Chen [1]   Ye Shi [1 2]   Jingya Wang [1 †]

## Abstract

We study real-time audio-responsive character control as a deployment-faithful problem: strictly causal, bounded-latency streaming that must generate coherent full-body motion at interactive frame rates while the audio condition can change abruptly (tempo shifts, drops, or user edits). Prior music-to-motion systems are largely optimized for offline generation with global context, and degrade in streaming rollouts where conditioning history becomes stale or unreliable. We introduce DiscoForcing, a streaming audio-driven diffusion framework that combines a causal music encoder that captures rhythmic structure and phase dynamics with a diffusion-forcing sequence model trained under heterogeneous noise levels across the temporal horizon. Building on this, we design a hybrid temporal schedule and a history-guided streaming sampler to explicitly trade off responsiveness against long-horizon consistency under non-stationary audio. Implemented in an end-to-end real-time interactive system with online avatar playback and humanoid deployment workflows, DiscoForcing delivers more stable long-horizon rollouts and sharper audio–motion alignment than prior baselines under matched causality and latency constraints while maintaining real-time throughput. Project Page: https://discoforcing.github.io/

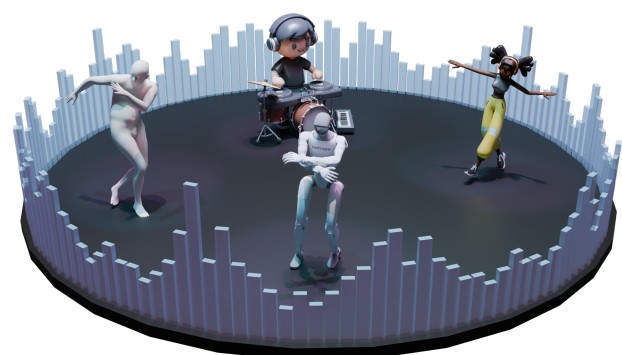

*Figure 1.* We introduce **DiscoForcing**, a real-time, audio-responsive character control system. Given online streaming audio inputs, DiscoForcing causally synthesizes continuous full-body motions in real time. The generated motion supports two deployment settings: (i) *avatar interactive control* for responsive animation and visualization, and (ii) *physics-based humanoid platform* by converting the predicted motion into executable humanoid joint commands.

## 1. Introduction

Real-time audio-responsive character control is a core capability for interactive embodied applications, spanning virtual reality avatars, animation-driven simulation, and physical humanoid deployment. In these settings, a system must convert a live music stream into continuous full-body motion that is simultaneously (i) *responsive* to newly arriving audio cues and user edits, (ii) *temporally coherent* over long rollouts, and (iii) *computationally feasible* at interactive frame rates under a strict per-frame budget. Beyond entertainment, this problem provides a concrete and deployment-faithful testbed for studying streaming generative policies under causality and latency constraints, which aligns with the broader goal of making generative sequence models usable in real interactive loops.

Despite rapid progress in music-to-motion generation, most prior methods are developed and validated in an offline regime. They commonly assume access to future audio context (Alexanderson et al., 2023; Li et al., 2021), rely on long temporal windows (Siyao et al., 2022), or perform non-causal inference that can revisit and refine past decisions (Li et al., 2024b). While these choices can improve offline metrics on fixed clips, they do not translate cleanly to streaming deployment. When executed online with zero lookahead, models often exhibit delayed reactions (Xiao et al., 2025), weakened beat synchronization (Tseng et al., 2023), and accumulating artifacts in long-horizon rollouts (Holden et al.,

*Equal contribution   [1]ShanghaiTech University   [2]InstAdapt.   Correspondence to: Jingya Wang <wangjingya@shanghaitech.edu.cn>.

*Proceedings of the 43rd International Conference on Machine Learning*, Seoul, South Korea. PMLR 306, 2026. Copyright 2026 by the author(s).

*Table 1.* Comparative analysis of DiscoForcing versus existing audio-driven human motion synthesis approaches.

| Method | online system | long-horizon | music transition | physical platform | user-interactive |
|---|---|---|---|---|---|
| Bailando (Siyao et al., 2022) | | ✓ | | | |
| EDGE (Tseng et al., 2023) | | ✓ | ✓ | | |
| Lodge (Li et al., 2024b) | | ✓ | | | |
| MEGADance (Yang et al., 2025a) | | | ✓ | | |
| FlowerDance (Yang et al., 2025b) | | | | | ✓ |
| DiscoForcing | ✓ | ✓ | ✓ | ✓ | ✓ |

2017). Importantly, this mismatch is structural rather than incidental: streaming execution changes the learning and inference problem because the model must act causally on incomplete evidence, commit to outputs without revision, and satisfy a hard real-time compute budget (Zhao et al., 2025).

Streaming audio-responsive control is challenging for three intertwined reasons. First, causality and bounded latency remove the anticipatory cues that offline systems exploit for beat alignment and phrase-level structure, making instantaneous synchronization substantially harder (Siyao et al., 2022; Barquero et al., 2024). Second, music streams are non-stationary: tempo shifts, drops, style switches, and user-driven edits can induce abrupt conditioning changes. Over-trusting motion history improves smoothness but slows reaction, whereas aggressively overriding history increases responsiveness but risks discontinuities and jitter (Fan et al., 2025; Huang et al., 2024). Third, long-horizon streaming amplifies error accumulation. Any small prediction error contaminates the future conditioning history, inducing a train–test distribution shift that is exacerbated when the audio condition itself evolves over time (Xiao et al., 2025; Holden et al., 2017).

A central limitation of the current literature is the lack of a **deployment-faithful end-to-end system** for audio-responsive character control. Most prior work is presented as an offline generation model evaluated on fixed clips, rather than a real-time interactive stack that users can directly experience. In practical use, however, the model must run online: it must process streaming audio, produce motion continuously in real time, and commit to outputs without retroactive revision (Dai et al., 2024). Without an integrated system that couples perception/encoding, streaming generation, and online execution, it remains unclear how existing models behave when placed in an actual interactive loop, and what design choices are necessary to deliver stable and responsive user-facing control.

In this work, we study real-time audio-responsive character control under explicit streaming constraints. We formulate the task as bounded-latency streaming motion generation: at each time step, the model receives a causal music representa-tion extracted from the live audio stream and a finite motion history buffer, and then predicts the next short motion chunk at interactive rate. This formulation mirrors practical interactive systems and enables principled, apples-to-apples evaluation under matched constraints.

Building on this formulation, we propose **DiscoForcing**, a streaming audio-driven diffusion framework that addresses the stability–responsiveness trade-off in non-stationary roll-outs. DiscoForcing couples (i) a causal music encoder that extracts rhythmic structure and phase dynamics from a sliding audio window with (ii) a diffusion-forcing sequence model trained with heterogeneous noise across time, improving robustness to imperfect or stale autoregressive histories. During inference, we use a streaming sampler with hybrid temporal scheduling, where temporal guidance combines a stable history-based prediction with an audio-conditioned prediction that preserves recent context while discounting distant history, thereby reducing lock-in to stale autoregressive states and improving responsiveness to incoming audio.

To close the evaluation gap, we also build an end-to-end real-time online verification system that runs the full loop under strict causality and low-latency constraints. The system enables real-time, audio-driven full-body motion generation from streaming audio inputs and supports two deployment settings: (i) a user-facing avatar platform for responsive animation and interaction, and (ii) a physics-based humanoid pipeline that executes the generated motion as joint-level control commands. This setup enforces matched causality and latency budgets and supports both interactive demos and embodied execution.

Under streaming settings, DiscoForcing improves both offline metrics and online rollout behavior compared with strong autoregressive and diffusion baselines. In particular, it yields better responsiveness to abrupt music changes without sacrificing continuity, while maintaining real-time throughput under strict per-frame computation budgets. These results indicate that diffusion-forcing models, when paired with deployment-aware conditioning management and streaming evaluation, offer a practical path toward robust audio-responsive character control in interactive applications, as shown in Table 1.

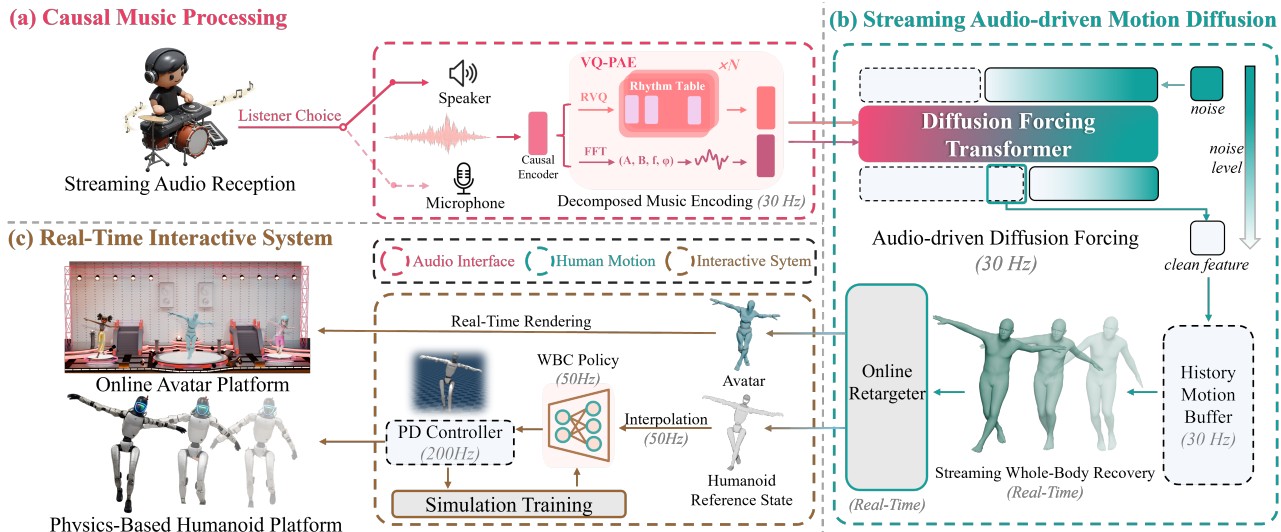

*Figure 2.* **System Pipeline.** DiscoForcing encodes live audio into a *causal* music feature (30 Hz) and generates continuous full-body motion via a diffusion-forcing transformer conditioned on the feature and a history buffer (30 Hz). The resulting motion is delivered to (i) an online avatar platform for retargeting and interactive Unity visualization, and (ii) a humanoid deployment stack that performs IK/interpolation and executes whole-body control with low-level PD tracking.

Our contributions are threefold:

- We present **DiscoForcing**, to our knowledge, the first end-to-end real-time audio-responsive character control framework that integrates causal audio encoding with streaming dance generation.

- We propose a latent diffusion-forcing model with decomposed music conditioning on rhythmic and periodic features, and introduce a temporal-guided strategy to balance fast reaction with long-horizon stability.

- We build a reproducible communication-based interactive system that enables real-time user interaction for both online avatar playback and a physics-based humanoid platform, supporting practical demonstrations and cross-platform validation within a unified runtime.

## 2. Related Works

**Audio-Driven Human Motion Synthesis.** Human motion synthesis has rapidly advanced with deep generative models that support rich conditioning signals, from action labels (Guo et al., 2020; Petrovich et al., 2021), text descriptions (Tevet et al., 2023; Chen et al., 2023; Zhang et al., 2023; Jiang et al., 2023; Wang et al., 2025b; Liu et al., 2025b) and various interactions (Ji et al., 2025; Tang et al., 2024; Deng et al., 2026) to audio. Audio-driven generation introduces a distinctive synchronization challenge: the model must align kinematic events to fine-grained rhythm while preserving coherent, natural full-body dynamics over long horizons. Early systems such as FACT (Li et al., 2021)

and Bailando (Siyao et al., 2022) use autoregressive transformers, but often accumulate errors in long rollouts, degrading beat timing and motion fidelity. Diffusion-based methods improve coverage of complex choreography and stylistic diversity (Alexanderson et al., 2023; Tseng et al., 2023; Li et al., 2023), with further advances in structure and controllability via coarse-to-fine sampling (Lodge (Li et al., 2024b)), bidirectional context (BADM (Zhang et al., 2024a)), and genre-aware experts (MEGADance (Yang et al., 2025a)). However, most approaches remain offline and non-causal, relying on future audio or full-track access for global consistency, which limits streaming use where the system must react causally to live audio and user edits under tight latency budgets.

**Real-Time Interactive Character Control.** Interactive character control is a core problem in animation and physics-based simulation, enabling users to steer virtual agents in a responsive and physically plausible way. Early approaches such as DeepPhase (Starke et al., 2022) and PFNN (Holden et al., 2017) rely on low-dimensional controls (e.g., heading direction, speed, simple style flags) to drive motion generation, yielding smooth locomotion but limited expressivity. Later methods like PADL (Juravsky et al., 2022) and AnySkill (Cui et al., 2024) use natural language as the control interface, formulating character control as language-conditioned motion generation. Building on this, CAMDM (Chen et al., 2024b), DART (Zhao et al., 2025) and FloodDiffusion (Cai et al., 2025) employ autoregressive diffusion models for text-driven motion synthesis, producing long-horizon motions with richer multimodality

and temporal coherence. In parallel, CLoSD (Tevet et al., 2025) explicitly closes the loop between motion generation and simulation to enforce physical feasibility, while recent work such as Diffuse-CLoC (Huang et al., 2025b) and Uni-Phys (Wu et al., 2025) jointly models state and action distributions to more tightly couple generative priors with control dynamics. However, audio-driven real-time interactive character control remains largely unexplored: music is highly non-stationary and diverse in rhythm and style, and existing systems rarely support a user-facing, real-time interaction and validation platform for evaluating control quality from the end-user's perspective.

**Diffusion Models for Sequence Modeling.** Diffusion probabilistic models (Sohl-Dickstein et al., 2015; Ho et al., 2020) invert a progressive noising process and have become dominant in image generation (Dhariwal & Nichol, 2021; Saharia et al., 2022; Rombach et al., 2022), motivating adaptations to sequential domains. A common approach is full-sequence denoising, jointly refining all timesteps (Ho et al., 2022; Janner et al., 2022; Song et al., 2023; Gupta et al., 2024). While effective for fixed-length synthesis, such non-causal formulations are ill-suited to variable-length or low-latency streaming, where outputs must be emitted incrementally. Several works combine diffusion with autoregressive factorization for incremental generation (Rasul et al., 2021; Wu et al., 2023; Kim et al., 2024; Ruhe et al., 2024), but naive AR execution can reintroduce exposure bias and compounding errors through history. Diffusion Forcing (Chen et al., 2024a) addresses this by training with token-wise heterogeneous noise, improving robustness to imperfect histories. Follow-ups refine conditioning management: History-Guided Diffusion (Song et al., 2025) stabilizes coherence with corrupted history, Self Forcing (Huang et al., 2025a) reduces train–test mismatch via self-generated history, and Rolling Forcing (Liu et al., 2025a) targets real-time generation with rolling-window denoising. Nevertheless, most streaming diffusion systems assume static or slowly varying global conditions, and do not directly address interactive control with non-stationary signals such as streaming music, which requires balancing continuity and responsiveness.

## 3. Methods

As shown in Fig. 2, our method is organized around three tightly coupled components for deployment-faithful streaming control: (i) **Causal Music Processing** that converts live audio into synchronized, strictly causal conditioning features; (ii) a **Streaming Audio-driven Motion Diffusion** model that performs bounded-latency, autoregressive diffusion generation over motion tokens/frames while remaining robust to imperfect histories; and (iii) a **Real-time Interactive System** that connects audio ingestion, online sampling, and avatar/robot playback in a closed timing loop with ex-plicit latency budgets.

### 3.1. Causal Music Processing

**Decomposed Music Conditioning.** To support strictly streaming control without lookahead, we introduce a causal music encoding pipeline that extracts a compact feature $c_t$ from a fixed-length sliding audio buffer at each step $t$. This enables the motion generator to respond immediately to newly arriving musical cues and user edits. Concretely, we adopt a Vector-Quantized Periodic Autoencoder (VQ-PAE) (Starke et al., 2022; Li et al., 2024a) to factorize music conditioning into discrete rhythmic tokens and continuous phase-alignment dynamics, producing an embedding that is simultaneously beat-aware and temporally smooth—well-suited for low-latency, online motion synthesis. At streaming step $t$, our generator is conditioned on a causal music feature

$$\mathbf{c}_t = [\mathbf{f}_t^{\mathrm{vq}};\ \mathbf{f}_t^{\mathrm{pae}}], \tag{1}$$

which is extracted from a fixed-length sliding audio window. The feature combines (i) a discrete rhythmic code via residual vector quantization and (ii) a continuous periodic alignment feature parameterized in the frequency domain.

**Discrete Rhythmic Encoding.** Let $\mathbf{w}_t$ be the waveform segment in the sliding buffer at step $t$. We compute a shared causal latent using a dilated 1D convolutional encoder:

$$\mathbf{f}^{\mathrm{causal}} = \mathrm{Conv1D}(\mathbf{w}_t), \tag{2}$$

where the dilation schedule is chosen to capture multi-scale temporal patterns while preserving causality. Then we feed the shared latent into a residual vector quantizer (RVQ) (Van Den Oord et al., 2017; Lee et al., 2022):

$$\mathbf{f}^{\mathrm{vq}} = \mathrm{RVQ}\big(\mathrm{FFN}_{\mathrm{vq}}(\mathbf{f}^{\mathrm{causal}})\big). \tag{3}$$

This branch captures discrete rhythmic patterns and high-level musical structure that are useful for motion triggering and style consistency.

**Periodic Alignment Encoding.** To obtain a smooth temporal alignment signal, we estimate frequency-domain parameters and predict the phase by a lightweight network using Fast Fourier Transformation (FFT) (Starke et al., 2022; Li et al., 2024a):

$$[\mathbf{A}, \mathbf{B}, \mathbf{F}] = \mathrm{FFT}(\mathbf{f}^{\mathrm{causal}}),\ \phi = \tan^{-1}\big(\mathrm{FC}_{\mathrm{phase}}(\mathbf{f}^{\mathrm{causal}})\big). \tag{4}$$

We reconstruct a periodic alignment feature in the time domain as:

$$\mathbf{f}^{\mathrm{pae}}(t) = \mathbf{A} \cdot \sin\big(2\pi(\mathbf{F} \cdot t - \phi)\big) + \mathbf{B}, \tag{5}$$

where $t$ denotes time within the window. Intuitively, $\phi$ controls the alignment (phase), while $\mathbf{A}$ and $\mathbf{B}$ control amplitude and offset for smooth transitions.

## 3.2. Streaming Audio-driven Motion Diffusion

We formulate real-time, audio-responsive interactive character control as a *streaming* motion generation problem. At the current time step $t$, our goal is to model the conditional distribution over the next $n$ motion frames, $\mathbf{m}_{t:t+n-1}$, given a historical motion buffer $\mathbf{m}_{t-h:t-1}$ and the synchronized music context $\mathbf{c}_t$.

**Streaming Motion Representation.** Prior dance generation methods (Tseng et al., 2023; Li et al., 2024b) often model motion in global, non-canonical coordinates. In autoregressive (AR) rollouts, global-frame error accumulation shifts the conditioning distribution, degrading long-horizon stability (e.g., drift and jitter). We thus adopt a *canonicalized incremental* representation that expresses dynamics in the root frame with velocity-like increments, yielding more stationary AR conditioning.

Many streaming generators (Tevet et al., 2025; Chen et al., 2024b) use the 263-D HumanML3D (Guo et al., 2022) feature, which recovers joint positions but not the joint rotations needed for real-time retargeting, requiring costly post-processing (Bogo et al., 2016). To remove this bottleneck, we follow (Xiao et al., 2025) and encode each frame as a 272-D vector:

$$\mathbf{m}_t = \left\{ \dot{r}_t^x, \dot{r}_t^z, \dot{r}_t^a, \mathbf{j}_t^p, \mathbf{j}_t^v, \mathbf{j}_t^r \right\} \in \mathbb{R}^{272}, \qquad (6)$$

where $\dot{r}_t^x, \dot{r}_t^z \in \mathbb{R}$ are root XZ-plane linear velocities, $\dot{r}_t^a \in \mathbb{R}^6$ is the 6D root angular velocity (Zhou et al., 2019), and $\mathbf{j}_t^p \in \mathbb{R}^{3K}$, $\mathbf{j}_t^v \in \mathbb{R}^{3K}$, $\mathbf{j}_t^r \in \mathbb{R}^{6K}$ are local joint positions, velocities, and rotations in the root frame. For SMPL (Loper et al., 2015), $K = 22$. This enables direct real-time reconstruction via forward kinematics, making AR streaming outputs immediately usable for low-latency retargeting without per-frame fitting.

**Motion Primitive Learning.** Given the strict low-latency requirements of streaming audio-driven motion generation, directly modeling high-dimensional raw motion dynamics is computationally prohibitive. To address this, we employ a motion Variational Autoencoder (VAE) (Kingma & Welling, 2014) to project the motion stream into a compact latent manifold, aligned with prior frameworks (Chen et al., 2023; Zhao et al., 2025). This strategy not only reduces the computational burden for the autoregressive backbone but also acts as an implicit low-pass filter, effectively removing high-frequency sensor jitter. Formally, let $\mathcal{T} = \{1, \ldots, T\}$ denote the indices of the full motion sequence. The VAE consists of an encoder $\mathcal{E}$ and a decoder $\mathcal{D}$, which map the processed motion stream $\mathbf{m}_\mathcal{T} \in \mathbb{R}^{T \times 272}$ to a latent sequence $\mathbf{z}_\mathcal{T} \in \mathbb{R}^{T \times D_z}$ and back to an approximation $\hat{\mathbf{m}}_\mathcal{T}$, respectively:

$$\mathbf{z}_\mathcal{T} = \mathcal{E}(\mathbf{m}_\mathcal{T}), \quad \hat{\mathbf{m}}_\mathcal{T} = \mathcal{D}(\mathbf{z}_\mathcal{T}). \qquad (7)$$

Our VAE is trained with an $\ell_2$ reconstruction loss to preserve motion fidelity and a Kullback-Leibler (KL) divergence loss to regularize the latent space towards a standard Gaussian prior:

$$\begin{aligned} \mathcal{L}_{\text{VAE}} = \ & \|\mathbf{m}_\mathcal{T} - \hat{\mathbf{m}}_\mathcal{T}\|_2^2 \\ & + \lambda D_{\text{KL}}\big( q(\mathbf{z}_\mathcal{T}|\mathbf{m}_\mathcal{T}) \parallel \mathcal{N}(\mathbf{0}, \mathbf{I}) \big), \end{aligned} \qquad (8)$$

where $\lambda$ is a weighting coefficient.

**Latent Diffusion Forcing.** To enable streaming generation with flexible temporal control, we adapt the diffusion forcing framework (Chen et al., 2024a) to our latent sequence modeling. We define a probability path specific to each temporal sequence index $t$ that transitions from clean data $\mathbf{x}_t^0 \equiv \mathbf{z}_t \sim q_{data}$ to pure noise $\mathbf{x}_t^1 \sim \mathcal{N}(\mathbf{0}, \mathbf{I})$ over a diffusion horizon $k_t \in [0, 1]$:

$$p(\mathbf{x}_t^{k_t} \mid \mathbf{x}_t^0) = \mathcal{N}(\mathbf{x}_t^{k_t}; \alpha_{k_t}\mathbf{x}_t^0, \sigma_{k_t}^2 \mathbf{I}), \qquad (9)$$

where $\alpha_{k_t}$ and $\sigma_{k_t}$ are predefined noise schedules satisfying the boundary conditions: $\alpha_0 = 1, \sigma_0 = 0$ and $\alpha_1 = 0, \sigma_1 = 1$. This formulation allows each token at sequence position $t$ to undergo denoising at an independent noise level determined by $k_t$. To reverse this process, based on the flow-based paradigm (Lipman et al., 2023; Li & He, 2025), we train a network $\mathbf{v}_\theta(\mathbf{x}_\mathcal{T}^{k_\mathcal{T}}, k_\mathcal{T}, \mathbf{c})$ to predict the conditional vector field $\mathbf{v}_\mathcal{T}$. Each component $\mathbf{v}_t$ of this field corresponds to the instantaneous velocity of the probability flow with respect to the diffusion time:

$$\mathbf{v}_t = \dot{\alpha}_{k_t}\mathbf{x}_t^0 + \dot{\sigma}_{k_t}\boldsymbol{\epsilon}_t, \qquad (10)$$

where $\dot{\alpha}_{k_t}$ and $\dot{\sigma}_{k_t}$ denote the derivatives of the schedule coefficients with respect to $k_t$, and $\boldsymbol{\epsilon}_t \sim \mathcal{N}(\mathbf{0}, \mathbf{I})$. The training objective minimizes the flow matching loss, computed selectively on noised tokens:

$$\mathcal{L}_{\text{DF}} = \mathbb{E}_{k_\mathcal{T}, \mathbf{z}_\mathcal{T}, \boldsymbol{\epsilon}_\mathcal{T}} \left[ \|\mathbf{v}_\theta(\mathbf{x}_\mathcal{T}^{k_\mathcal{T}}, k_\mathcal{T}, \mathbf{c}) - \mathbf{v}_\mathcal{T}\|_\mathcal{K}^2 \right], \qquad (11)$$

where the norm $\|\cdot\|_\mathcal{K}$ (defined as $\|\mathbf{u}\|_\mathcal{K}^2 = \sum_{t:k_t>0} \|u_t\|^2$) serves as a masking mechanism. Since tokens with $k_t = 0$ remain in the clean state and do not require denoising, this mask ensures the model focuses exclusively on learning the active transport dynamics.

**Hybrid Temporal Schedule.** Distinct from vanilla Diffusion Forcing which relies solely on random noise schedules, we employ a hybrid strategy to bridge the gap between training and streaming inference, while also encouraging consistent denoising patterns. At each iteration, as detailed in Algorithm 2, we stochastically sample a noise schedule type from a categorical distribution defined over three variants:

1) **Random Schedule**: $k_t^{\text{rand}} \sim \mathcal{U}(0,1)$. This independent sampling ensures the model retains the flexibility to handle arbitrary noise states.

2) **Monotonic Schedule**. Motivated by (Zhang et al., 2024b; Liu et al., 2025a), we define a schedule based on a randomly sampled current time step $\tau \in [0, T]$ and a window size $l$:

$$k_t^{\text{mono}} = \text{clamp}\left(\frac{t - (\tau - l)}{l}, 0, 1\right). \qquad (12)$$

Under this schedule, frames before $\tau - l$ are clean ($k_t = 0$), frames after $\tau$ are pure noise ($k_t = 1$), and the intermediate window transitions linearly.

3) **Trapezoid Schedule**. Following (Chen et al., 2024a; Song et al., 2025), we further introduce a trapezoidal noise profile that assigns higher noise levels to tokens significantly preceding the current time step $\tau$:

$$k_t^{\text{hist}} = \text{clamp}\left(\frac{(\tau - l_{\text{ctx}} - l) - t}{l_{\text{hist}}}, 0, 1\right),$$
$$k_t^{\text{trap}} = \max(k_t^{\text{hist}}, k_t^{\text{mono}}). \qquad (13)$$

Here, $l_{\text{ctx}}$ denotes the length of the clean context window and $l_{\text{hist}} > 0$ determines the rate at which noise levels increase for the distant past.

**Streaming Inference with Temporal Guidance.** We perform bounded-latency streaming generation by maintaining a fixed-length FIFO denoising window of size $l$ at the tail of the sequence (Kim et al., 2024), and apply hybrid noise schedule as temporal guidance to denoise all tokens in the window jointly (Alg. 1). Once the leftmost token in the window reaches the clean state, we emit it as the next streaming output token, then shift the buffer by one position and append a new token initialized as pure Gaussian noise, continuing the streaming process.

Inspired by (Ho & Salimans, 2021; Song et al., 2025), we employ temporal guidance to mitigate long-term motion drift. By corrupting the distant history with a trapezoid noise schedule, we implicitly filter out high-frequency priors that may contradict current control signals. The guided velocity is then computed as a weighted sum with scale $\omega$:

$$\mathbf{v}_{\mathcal{T}}^{\text{guided}} = \mathbf{v}_\theta(\hat{\mathbf{x}}_{\mathcal{T}}^{k_{\mathcal{T}}^{\text{mono}}}, k_{\mathcal{T}}^{\text{mono}}, \varnothing)$$
$$+ \omega[\mathbf{v}_\theta(\hat{\mathbf{x}}_{\mathcal{T}}^{k_{\mathcal{T}}^{\text{trap}}}, k_{\mathcal{T}}^{\text{trap}}, \mathbf{c}) - \mathbf{v}_\theta(\hat{\mathbf{x}}_{\mathcal{T}}^{k_{\mathcal{T}}^{\text{mono}}}, k_{\mathcal{T}}^{\text{mono}}, \varnothing)]. \qquad (14)$$

### 3.3. Real-Time Interactive System

To validate our streaming formulation beyond offline metrics, we build an end-to-end real-time interactive system that supports both virtual and physical deployment. As illustrated in Fig. 2, the system exposes two application frontends: (i) an **online avatar interaction platform** in Unity for

---

**Algorithm 1** Streaming Inference with Temporal Guidance

**Input:** Audio stream $\mathbf{c}$, Model $\mathbf{v}_\theta$, Max length $T$, Window size $l$, Guidance scale $\omega$, Schedule params $l_{\text{ctx}}, l_{\text{hist}}$.
**Initialize:** Latent buffer $\hat{\mathbf{x}} = \varnothing$, Global step $\tau \leftarrow 0$, Solver step $\delta \leftarrow 1/l$.
        ▷ ensures one denoising step per output token
**while** streaming **do**
  $\tau \leftarrow \tau + 1$.
  $\hat{\mathbf{x}} \leftarrow [\hat{\mathbf{x}}, \boldsymbol{\epsilon}]$, where $\boldsymbol{\epsilon} \sim \mathcal{N}(\mathbf{0}, \mathbf{I})$.
  $\hat{\mathbf{x}} \leftarrow \hat{\mathbf{x}}[-T :]$.         ▷ maintain the buffer
  Define active window $\mathcal{W} \leftarrow \{\tau - l + 1, \ldots, \tau\}$.
  Define history indices $\mathcal{H} \leftarrow \{\text{indices before } \mathcal{W}\}$.
  $k_t^{\text{mono}} \leftarrow \Phi_{\text{mono}}(t, \tau, l)$ for $t \in \mathcal{W}$.
  $k_t^{\text{trap}} \leftarrow \Phi_{\text{trap}}(t, \tau, l, l_{\text{ctx}}, l_{\text{hist}})$ for all $t$.
        ▷ construct the schedule
  $\hat{\mathbf{x}}^{\text{trap}} \leftarrow \hat{\mathbf{x}}$; Sample $\boldsymbol{\epsilon}' \sim \mathcal{N}(\mathbf{0}, \mathbf{I})$.
  $\hat{\mathbf{x}}_{\mathcal{H}}^{\text{trap}} \leftarrow \alpha(k_{\mathcal{H}}^{\text{trap}}) \odot \hat{\mathbf{x}}_{\mathcal{H}} + \sigma(k_{\mathcal{H}}^{\text{trap}}) \odot \boldsymbol{\epsilon}_{\mathcal{H}}'$.
        ▷ here $\odot$ denotes element-wise multiplication
  $\mathbf{v}_{\text{cond}} \leftarrow \mathbf{v}_\theta(\hat{\mathbf{x}}^{\text{trap}}, k^{\text{trap}}, \mathbf{c})$.
  $\mathbf{v}_{\text{hist}} \leftarrow \mathbf{v}_\theta(\hat{\mathbf{x}}, k^{\text{mono}}, \varnothing)$.
  $\mathbf{v} \leftarrow \mathbf{v}_{\text{hist}} + \omega(\mathbf{v}_{\text{cond}} - \mathbf{v}_{\text{hist}})$.   ▷ temporal guidance
  $\hat{\mathbf{x}}_{\mathcal{W}} \leftarrow \hat{\mathbf{x}}_{\mathcal{W}} - \mathbf{v}_{\mathcal{W}} \cdot \delta$.      ▷ denoise the window
  Emit token $\hat{\mathbf{x}}_{\tau - l + 1}$ as output $\hat{\mathbf{z}}$.
**end while**

---

user-facing animation interaction and (ii) a **physics-based humanoid deployment pipeline** targeting the Unitree G1 humanoid robot for embodied execution. The streaming generator communicates with both Unity and the G1 robot exclusively via ROS2.

**Online Avatar Platform.** We implement an interactive animation and visualization platform in Unity, where virtual characters are controlled using SMPL parameters (Loper et al., 2015). The streaming generator transmits real-time motion data to Unity through ROS2, enabling the characters to perform synchronized movements with the input music. This setup allows for live, user-facing evaluation of the generated motions under realistic streaming conditions.

**Physics-Based Humanoid Platform.** We further validate the streaming generator on a physical humanoid robot. SMPL-format motions produced by the streaming generator are first retargeted to the Unitree G1 using GMR (Araujo et al., 2025). The resulting reference trajectories are subsequently tracked by a whole-body control (WBC) policy trained on large-scale motion datasets in simulation (Liao et al., 2025; Mahmood et al., 2019; Punnakkal et al., 2021). To ensure temporal alignment, the retargeted motion sequences are interpolated to match the tracker's 50Hz control frequency. All of these steps are performed in real time, enabling the robot to dynamically adapt its dance movements in response to continuously changing music.

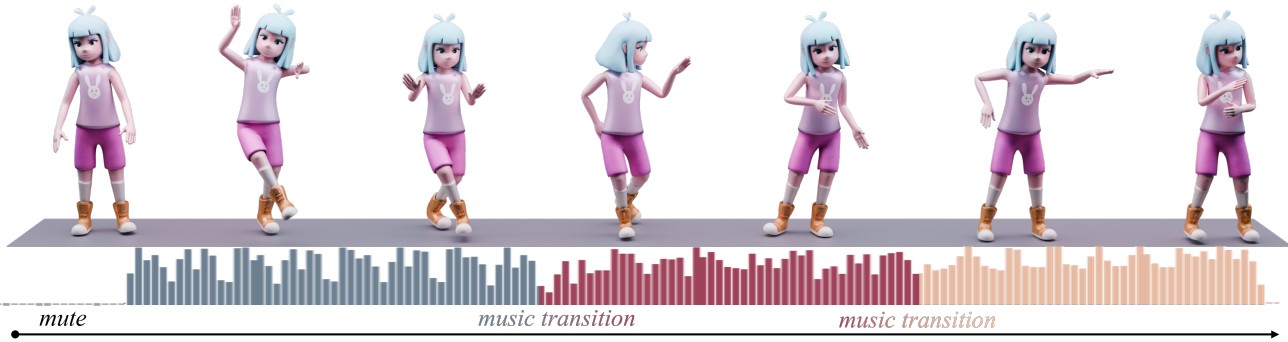

*Figure 3.* **Demonstration of Our Method.** In a strictly causal, bounded-latency online streaming rollout, DiscoForcing keeps the character stationary during silent segments (mute), and immediately generates beat-synchronized full-body dance once music resumes. As the input stream undergoes multiple music transitions, our model adapts in real time to the changing audio while maintaining long-horizon temporal coherence and smooth motion continuity.

*Table 2.* Quantitative comparison on FineDance (Li et al., 2023) dataset. GT stands for ground truth, A higher or lower value is better for ↑ or ↓, and → means the value closer to ground truth is better.

| | $FID_k\downarrow$ | $FID_g\downarrow$ | FSR↓ | $Div_k\rightarrow$ | $Div_g\rightarrow$ | BAS↑ |
|---|---|---|---|---|---|---|
| GT | – | – | 0.062 | 9.94 | 7.54 | 0.201 |
| FACT | 113.38 | 97.05 | 0.284 | 3.36 | **6.37** | 0.183 |
| Bailando | 82.81 | 28.17 | 0.188 | 7.74 | 6.25 | 0.202 |
| EDGE | 94.34 | 50.38 | 0.200 | **8.13** | 6.45 | 0.212 |
| Lodge | 50.00 | 35.52 | **0.028** | 5.67 | 4.96 | **0.226** |
| MEGA | 50.00 | 13.02 | 0.243 | 6.23 | 6.27 | **0.226** |
| **DiscoForcing** | **23.84** | **8.62** | 0.142 | 5.98 | 5.99 | 0.225 |

*Table 3.* Comparison on AIST++ (Li et al., 2021) dataset.

| | $FID_k\downarrow$ | $FID_g\downarrow$ | $Div_k\rightarrow$ | $Div_g\rightarrow$ | BAS↑ |
|---|---|---|---|---|---|
| Ground Truth | - | - | 8.19 | 7.45 | 0.237 |
| FACT | 35.35 | 22.11 | 5.94 | 6.18 | 0.221 |
| Bailando | 28.16 | **9.62** | **7.83** | **6.34** | 0.233 |
| EDGE | 42.16 | 22.12 | 3.96 | 4.61 | 0.233 |
| Lodge | 37.09 | 18.79 | 5.58 | 4.85 | 0.242 |
| MEGA | 25.89 | 12.62 | 5.84 | 6.23 | 0.238 |
| **DiscoForcing** | **18.87** | 11.57 | 6.76 | 6.31 | **0.244** |

## 4. Experiments

### 4.1. Experimental Setup

**Datasets.** To validate our method, we utilize two public dance datasets that provide 3D dance motions synchronized with accompanying music: FineDance (Li et al., 2023) and AIST++ (Li et al., 2021). **FineDance** stands as the largest publicly available dataset to date, providing 7.7 hours of music-paired optical motion capture data at 30 FPS. **AIST++** serves as a widely used standard benchmark, providing 5.2

hours of motion sequences reconstructed from multi-view videos at 60 FPS. For both datasets, we strictly follow the official data split for fair comparison and cut all examples to 30 FPS.

**Evaluation Metrics.** Following established protocols (Yang et al., 2025a; Li et al., 2024b; Tseng et al., 2023), we evaluate: **1) Motion Quality** (realism and physical plausibility), **2) Motion Diversity** (variety and non-repetitiveness), and **3) Music–Dance Correlation** (rhythmic synchronization). For Motion Quality, we use Fréchet Inception Distance (FID) (Heusel et al., 2017) to measure distributional similarity between generated and real motions. Following (Li et al., 2021), we report $\mathbf{FID}_k$ and $\mathbf{FID}_g$ on kinetic and geometric features, capturing physical smoothness and choreographic quality. We also adopt **FSR** (Li et al., 2024b) to penalize foot-skating via consistency between center-of-mass acceleration and foot velocity. For Motion Diversity, we compute the average pairwise Euclidean distance of generated motion features and report $\mathbf{Div}_k$ and $\mathbf{Div}_g$ in the kinetic and geometric spaces (Siyao et al., 2022). For Music–Dance Correlation, we use Beat Alignment Score (**BAS**) (Li et al., 2021) to quantify alignment between kinematic motion beats and musical audio beats.

### 4.2. Music-to-Dance Evaluation

**Baselines.** We compare our method against several representative music-to-dance approaches: FACT (Li et al., 2021), an autoregressive cross-modal transformer; Bailando (Siyao et al., 2022), a VQ-VAE with actor–critic GPT framework; EDGE (Tseng et al., 2023), a diffusion transformer conditioned on Jukebox audio features; Lodge (Li et al., 2024b), a coarse-to-fine diffusion model with primitive generation and refinement; and MEGADance (Yang et al., 2025a), a Mixture-of-Experts Mamba–Transformer

*Table 4.* Ablation studies of music-to-dance generation on AIST++ (Li et al., 2021) dataset, where underlined denotes the design we choose.

| Class | Settings | Motion Quality | | | Motion Diversity | | BAS↑ | Latency |
|---|---|---|---|---|---|---|---|---|
| | | $FID_k\downarrow$ | $FID_g\downarrow$ | FSR↓ | $Div_k\rightarrow$ | $Div_g\rightarrow$ | | (ms / frame) |
| Ground Truth | | – | – | 0.007 | 8.19 | 7.45 | 0.237 | – |
| Representation | 151d | 27.83 | 15.78 | 0.201 | 5.57 | 6.62 | 0.233 | 25.65 |
| | 263d | 26.47 | 8.30 | 0.060 | 8.53 | 7.62 | 0.245 | 26.60 |
| | 272d | 25.49 | 13.30 | 0.115 | 10.37 | 7.66 | 0.247 | 26.68 |
| Music Encoder | Librosa | 25.49 | 13.30 | 0.115 | 10.37 | 7.66 | 0.247 | 26.68 |
| | VQ-PAE | 23.23 | 12.28 | 0.097 | 6.50 | 6.68 | 0.238 | 26.73 |
| Guidance | CFG | 23.23 | 12.28 | 0.097 | 6.50 | 6.68 | 0.238 | 26.73 |
| | TG | 18.87 | 11.57 | 0.059 | 6.76 | 6.31 | 0.244 | 26.26 |
| Time-steps | 5 | 28.63 | 12.23 | 0.062 | 5.41 | 5.85 | 0.242 | 14.02 |
| | 10 | 18.87 | 11.57 | 0.059 | 6.76 | 6.31 | 0.244 | 26.26 |
| | 100 | 17.58 | 11.29 | 0.080 | 8.99 | 6.86 | 0.248 | 261.91 |
| DiscoForcing | | 18.87 | 11.57 | 0.059 | 6.76 | 6.31 | 0.244 | 26.26 |

hybrid.

**Comparisons.** As summarized in Table 2 and Table 3, DiscoForcing achieves the best overall motion quality on FineDance, with substantially lower $FID_k$ and $FID_g$ than all baselines while keeping competitive BAS and diversity; on AIST++, DiscoForcing further improves kinetic $FID_k$ over strong diffusion and MoE baselines, indicating better roll-out stability under matched streaming constraints. Notably, methods optimized for offline/global-context generation may achieve strong alignment or specific sub-metrics, but DiscoForcing provides a more balanced trade-off between long-horizon coherence and responsiveness in a deployment-faithful setting.

### 4.3. Ablation Study

**Motion Representation.** As shown in Table 4, we compare our 272-dim representation with the 263-dim HumanML3D format and a 151-dim baseline. While 263-dim offers slight geometric gains, it lacks rotations and requires costly inverse kinematics, violating real-time constraints. Our 272-dim design best balances quality and deployability, improving fidelity and beat alignment over 151-dim while enabling low-latency forward-kinematics reconstruction.

**Music Encoder.** Hand-crafted Librosa (McFee et al., 2015) features yield marginally better beat alignment due to explicit onset cues, but our learned VQ-PAE substantially improves realism. This indicates rule-based descriptors capture low-level pulses, whereas VQ-PAE encodes richer stylistic and semantic cues for natural control.

**Sampling Strategy.** We ablate inference settings. Temporal Guidance (TG) outperforms Classifier-Free Guidance (CFG) by suppressing autoregressive drift and improving synchronization. For compute, 10 denoising steps are optimal: 100 steps brings diminishing gains while exceeding real-time limits, and 5 steps harms plausibility. We therefore adopt 10-step denoising to sustain stable 30 FPS throughput.

## 5. Conclusion

We present DiscoForcing, a novel streaming framework for audio-driven interactive character control that operates under strict causality and low-latency constraints. The core contribution is an autoregressive diffusion forcing generator that conditions on a short motion history and synchronized causal music features, enabling long-horizon, real-time motion synthesis without requiring future context or offline planning. We further describe a practical end-to-end system pipeline that supports both online avatar visualization and humanoid deployment via standard retargeting and tracking control. Experiments demonstrate that our approach improves motion quality and temporal consistency under streaming rollouts, while remaining responsive to time-varying audio conditions such as rhythm changes and abrupt transitions. We hope this work serves as a step toward deployment-faithful generative sequence modeling for interactive embodied applications, and we expect future work to explore stronger robustness under distribution shift, richer user control signals, stronger physical feasibility constraints, broader real-world testing, and tighter integration with downstream controllers for safer and more reliable deployment.

## Acknowledgements

This work was supported by National Natural Science Foundation of China (62406195, 62303319), Shanghai Local College Capacity Building Program (23010503100), HPC Platform of ShanghaiTech University, Core Facility Platform of Computer Science and Communication of ShanghaiTech University, and MoE Key Laboratory of Intelligent Perception and Human-Machine Collaboration (ShanghaiTech University), and Shanghai Engineering Research Center of Intelligent Vision and Imaging.

## Impact Statement

This paper presents a streaming, audio-conditioned motion generation method for interactive character control. The work may benefit animation, virtual reality, and embodied AI research by enabling lower-latency, responsive motion synthesis. Potential risks include misuse for deceptive synthetic media when combined with identity-linked avatars, bias inherited from training data, and safety concerns if outputs are executed on physical robots without appropriate constraints. We recommend responsible deployment practices, including conservative control and safety limits for real hardware, careful dataset documentation, and appropriate handling of audio and licensing.

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

# A. Additional Methods

## A.1. Open-Source Statement

To foster reproducibility and facilitate follow-up research on real-time audio-responsive character control, we commit to releasing our codebase, configuration files, and pretrained checkpoints, as well as scripts for motion dataset processing, metric evaluation, and our ROS2-based system and web demonstration upon publication of this work, subject to dataset and hardware licensing constraints.

## A.2. Theoretical Foundation of Latent Diffusion Forcing

In this section, we provide the theoretical background for our method. We first introduce the general framework of Conditional Flow Matching and then derive its extension to sequence modeling via independent temporal factorization.

**Conditional Flow Matching.** We formulate motion generation using the Conditional Flow Matching (CFM) framework (Lipman et al., 2023), which learns a deterministic probability flow ODE to transform a standard Gaussian distribution into the data distribution. We define a conditional probability path $p_k(\mathbf{x} \mid \mathbf{x}^0)$ that interpolates between clean motion $\mathbf{x}^0$ and pure noise $\mathbf{x}^1 \sim \mathcal{N}(\mathbf{0}, \mathbf{I})$ over a continuous time horizon $k \in [0, 1]$. This process is parameterized by differentiable signal and noise schedules, $\alpha_k$ and $\sigma_k$, satisfying boundary conditions $\alpha_0 = 1, \sigma_0 = 0$ and $\alpha_1 = 0, \sigma_1 = 1$. An intermediate sample $\mathbf{x}^k$ is obtained via:

$$\mathbf{x}^k = \alpha_k \mathbf{x}^0 + \sigma_k \boldsymbol{\epsilon}, \quad \text{where } \boldsymbol{\epsilon} \sim \mathcal{N}(\mathbf{0}, \mathbf{I}). \tag{15}$$

This path is generated by a unique conditional vector field $\mathbf{u}_k(\mathbf{x}^k \mid \mathbf{x}^0)$ defined as the time derivative of the sample:

$$\mathbf{u}_k(\mathbf{x}^k \mid \mathbf{x}^0) = \frac{d\mathbf{x}^k}{dk} = \dot{\alpha}_k \mathbf{x}^0 + \dot{\sigma}_k \boldsymbol{\epsilon}. \tag{16}$$

The training objective is to regress a neural velocity estimator $\mathbf{v}_\theta(\mathbf{x}^k, k, \mathbf{c})$ to approximate this target field. We optimize $\theta$ by minimizing the flow matching loss over uniform time steps $k \sim \mathcal{U}[0, 1]$ and data samples:

$$\mathcal{L}_{\text{FM}} = \mathbb{E}_{k, \mathbf{x}^0, \boldsymbol{\epsilon}} \left[ \|\mathbf{v}_\theta(\mathbf{x}^k, k, \mathbf{c}) - (\dot{\alpha}_k \mathbf{x}^0 + \dot{\sigma}_k \boldsymbol{\epsilon})\|^2 \right]. \tag{17}$$

During inference, we generate motion by sampling noise $\mathbf{x}^1$ and numerically integrating the learned ODE $d\mathbf{x}/dk = \mathbf{v}_\theta(\mathbf{x}, k, \mathbf{c})$ backward from $k = 1$ to $0$.

**Independent Per-Token Diffusion** Standard diffusion models assume a global diffusion time $k$ shared across all dimensions. However, for streaming applications where history is fixed (clean) and the future is unknown (noisy), we require flexible control over the noise level of individual sequence elements. Following Diffusion Forcing (Chen et al., 2024a), we generalize the scalar time $k$ to a high-dimensional time vector $k_\mathcal{T} = \{k_1, \ldots, k_T\}$, where each token $\mathbf{x}_t$ in the sequence $\mathbf{x}_\mathcal{T}$ is assigned an independent diffusion time $k_t$.

The joint conditional probability path for the sequence factorizes into independent Gaussian paths:

$$p(\mathbf{x}_\mathcal{T}^{k_\mathcal{T}} \mid \mathbf{x}_\mathcal{T}^0) = \prod_{t=1}^{T} p(\mathbf{x}_t^{k_t} \mid \mathbf{x}_t^0) = \prod_{t=1}^{T} \mathcal{N}(\mathbf{x}_t^{k_t}; \alpha_{k_t} \mathbf{x}_t^0, \sigma_{k_t}^2 \mathbf{I}). \tag{18}$$

A key property of CFM is that if the conditional distribution factorizes, the target vector field decomposes element-wise. Specifically, the total derivative of the sequence state $\mathbf{x}_\mathcal{T}^{k_\mathcal{T}}$ with respect to the time-vector configuration $k_\mathcal{T} = \{k_1, \ldots, k_T\}$ simplifies to the collection of individual token velocities. We denote this target sequence vector field by

$$\mathbf{v}_\mathcal{T}^{\text{target}} \equiv \mathbf{u}_\mathcal{T}\left(\mathbf{x}_\mathcal{T}^{k_\mathcal{T}} \mid \mathbf{x}_\mathcal{T}^0\right) = \frac{d\mathbf{x}_\mathcal{T}^{k_\mathcal{T}}}{dk_\mathcal{T}} = \dot{\boldsymbol{\alpha}}_{k_\mathcal{T}} \odot \mathbf{x}_\mathcal{T}^0 + \dot{\boldsymbol{\sigma}}_{k_\mathcal{T}} \odot \boldsymbol{\epsilon}_\mathcal{T} = [\mathbf{v}_1, \ldots, \mathbf{v}_T], \tag{19}$$

where $\dot{\boldsymbol{\alpha}}_{k_\mathcal{T}} := [\dot{\alpha}_{k_1}, \ldots, \dot{\alpha}_{k_T}]$ and $\dot{\boldsymbol{\sigma}}_{k_\mathcal{T}} := [\dot{\sigma}_{k_1}, \ldots, \dot{\sigma}_{k_T}]$ apply element-wise. Hence, the target vector field for the $t$-th token depends only on its local time $k_t$:

$$\mathbf{v}_t = \mathbf{u}_t(\mathbf{x}_t^{k_t} \mid \mathbf{x}_t^0) = \dot{\alpha}_{k_t} \mathbf{x}_t^0 + \dot{\sigma}_{k_t} \boldsymbol{\epsilon}_t. \tag{20}$$

This decomposition allows us to train a single network $\mathbf{v}_\theta(\mathbf{x}_\mathcal{T}^{k_\mathcal{T}}, k_\mathcal{T}, \mathbf{c})$ that takes the heterogeneous sequence as input and predicts a sequence velocity field that minimizes the sum of squared errors across all tokens.

---

**Algorithm 2** Training of Latent Diffusion Forcing

---

1: **Input:** Dataset $\mathcal{D}$, Model $\mathbf{v}_\theta$, Schedule probabilities $\{\pi_{\text{rand}}, \pi_{\text{mono}}, \pi_{\text{trap}}\}$, Schedule params $\{l, l_{\text{ctx}}, l_{\text{hist}}\}$.
2: **while** not converged **do**
3:     *// 1. Sampling*
4:     Sample clean latents $\mathbf{z}_\mathcal{T} \sim \mathcal{D}$ and audio condition $\mathbf{c}$.
5:     Sample current virtual stream time $\tau \sim \mathcal{U}(0, |\mathcal{T}|)$.
6:     Sample schedule type $s \sim \text{Cat}(\pi_{\text{rand}}, \pi_{\text{mono}}, \pi_{\text{trap}})$.
7:     Sample noise $\boldsymbol{\epsilon}_\mathcal{T} \sim \mathcal{N}(\mathbf{0}, \mathbf{I})$.
8:     *// 2. Schedule Assignment (Hybrid Strategy)*
9:     **if** $s = \text{rand}$ **then**
10:         $k_t \sim \mathcal{U}(0, 1)$ for all $t \in \mathcal{T}$.
11:     **else if** $s = \text{mono}$ **then**
12:         $k_t \leftarrow \Phi_{\text{mono}}(t, \tau, l)$ for all $t \in \mathcal{T}$.
13:     **else if** $s = \text{trap}$ **then**
14:         $k_t \leftarrow \Phi_{\text{trap}}(t, \tau, l, l_{\text{ctx}}, l_{\text{hist}})$ for all $t \in \mathcal{T}$.
15:     **end if**
16:     Define active mask $\mathcal{M} = \{t \in \mathcal{T} \mid k_t > 0\}$.
17:     *// 3. Forward Process & Optimization*
18:     $\mathbf{x}_t \leftarrow \alpha(k_t)\mathbf{z}_t + \sigma(k_t)\boldsymbol{\epsilon}_t$.            ▷ Noisy State
19:     $\mathbf{v}_t^{\text{target}} \leftarrow \dot{\alpha}(k_t)\mathbf{z}_t + \dot{\sigma}(k_t)\boldsymbol{\epsilon}_t$.       ▷ Target Velocity
20:     $\hat{\mathbf{v}}_\mathcal{T} \leftarrow \mathbf{v}_\theta(\mathbf{x}_\mathcal{T}, k_\mathcal{T}, \mathbf{c})$.
21:     $\mathcal{L} \leftarrow \frac{1}{|\mathcal{M}|} \sum_{t \in \mathcal{M}} \|\hat{\mathbf{v}}_t - \mathbf{v}_t^{\text{target}}\|^2$.
22:     Update $\theta \leftarrow \theta - \eta\nabla_\theta\mathcal{L}$.
23: **end while**

---

### A.3. Detailed Diffusion Forcing Training and Inference Implementation

**Training Procedure.** Our training process, as outlines in **Algorithm 2**, aims to optimize the velocity estimation network $\mathbf{v}_\theta$ to handle both independent noise states and structured streaming trajectories. Each training step proceeds sequentially as follows:

**1) State Initialization.** Given a batch of clean motion latent sequences $\mathbf{z}_\mathcal{T}$ and audio conditions $\mathbf{c}$, we first sample a *virtual stream time* $\tau$ uniformly from $[0, T]$ for each sequence. This $\tau$ acts as a temporal anchor, simulating a snapshot of the streaming buffer at an arbitrary time point.

**2) Schedule Selection.** To determine the noise distribution pattern, we sample a schedule type $s$ from a categorical distribution parameterized by fixed probabilities $\Pi = [\pi_{\text{rand}}, \pi_{\text{mono}}, \pi_{\text{trap}}]$ such that $\sum \pi = 1$:

$$s \sim \text{Cat}(s \mid \Pi), \quad \text{where } s \in \{\text{rand}, \text{mono}, \text{trap}\}. \tag{21}$$

Subsequently, per-token noise levels $k_t$ are assigned based on the selected $s$: if $s = \text{rand}$, $k_t \sim \mathcal{U}(0, 1)$ i.i.d.; otherwise, $k_t$ is deterministically computed via the window-based functions $\Phi_{\text{mono}}(t, \tau)$ or $\Phi_{\text{trap}}(t, \tau)$ (as defined in (12), (13)), which impose a structured transition from clean history to noisy future around $\tau$.

**3) Forward Process and Objective.** Based on the defined probability path (9), we generate the noisy state inputs $\mathbf{x}_\mathcal{T}$ from the clean latents $\mathbf{z}_\mathcal{T}$ (where $\mathbf{x}_t^0 = \mathbf{z}_t$). Instead of sampling iteratively, we apply the reparameterization trick to sample $\mathbf{x}_t^{k_t}$ directly at the determined noise level $k_t$:

$$\mathbf{x}_t^{k_t} = \alpha_{k_t}\mathbf{x}_t^0 + \sigma_{k_t}\boldsymbol{\epsilon}_t, \quad \text{where } \boldsymbol{\epsilon}_t \sim \mathcal{N}(\mathbf{0}, \mathbf{I}). \tag{22}$$

Specifically, we adopt a *linear noise schedule* (Li & He, 2025) (i.e., $\alpha_{k_t} = 1 - k_t$ and $\sigma_{k_t} = k_t$), which corresponds to an optimal transport displacement path:

$$\mathbf{x}_t^{k_t} = (1 - k_t)\mathbf{x}_t^0 + k_t\boldsymbol{\epsilon}_t, \quad \text{where } \boldsymbol{\epsilon}_t \sim \mathcal{N}(\mathbf{0}, \mathbf{I}). \tag{23}$$

Accordingly, the target flow velocity field $\mathbf{v}_t^{\text{target}}$, defined as the time derivative of the state $\mathbf{x}_t$, simplifies to the difference

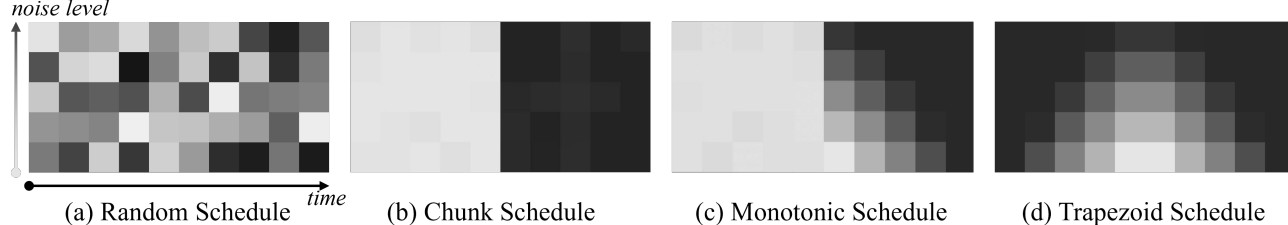

*noise level*

*time*

(a) Random Schedule      (b) Chunk Schedule      (c) Monotonic Schedule      (d) Trapezoid Schedule

*Figure 4.* **Hybrid temporal noise schedules for bounded-latency streaming diffusion.** Color encodes the per-timestep diffusion time (noise level) $k_t$. From left to right: an i.i.d. random schedule $k_t^{\text{rand}}$, a windowed monotonic schedule $k_t^{\text{mono}}$, a trapezoid (history-corrupting) schedule $k_t^{\text{trap}}$, and the resulting effective schedule used for temporal guidance. In contrast, chunk-based diffusion typically must fully denoise every frame in a chunk/window at each update, making computation scale with chunk length and inducing substantial latency; our hybrid/asynchronous scheduling reallocates denoising budget to the most critical region to preserve responsiveness while maintaining long-horizon stability.

between noise and data:

$$\mathbf{v}_t^{\text{target}} = \frac{\mathrm{d}\mathbf{x}_t^{k_t}}{\mathrm{d}k_t} = \boldsymbol{\epsilon}_t - \mathbf{x}_t^0. \tag{24}$$

To learn this vector field, we train the network $\mathbf{v}_\theta(\mathbf{x}_\mathcal{T}^{k_\mathcal{T}}, k_\mathcal{T}, \mathbf{c})$ by minimizing the mean squared error against $\mathbf{v}_t^{\text{target}}$. Crucially, to prevent wasting capacity on fully clean data, we apply a binary mask $\mathcal{M}_t = \mathbb{I}(k_t > 0)$ to the objective. This ensures the loss is calculated *exclusively* on active tokens, focusing the learned gradients solely on the transport dynamics within the denoising part.

**Alternative Parameterizations.** While our primary formulation trains the network to directly predict the velocity field $\mathbf{v}_\theta \approx \boldsymbol{\epsilon} - \mathbf{x}^0$, the Flow Matching framework is fully compatible with standard diffusion parameterizations, specifically data prediction ($\mathbf{x}^0$ - pred) and noise prediction ($\boldsymbol{\epsilon}$ - pred) (Li & He, 2025). Under the sequence-wise optimal transport path $\mathbf{x}^{k_\mathcal{T}} = (1 - k_\mathcal{T}) \odot \mathbf{x}^0 + k_\mathcal{T} \odot \boldsymbol{\epsilon}$, the velocity field relates analytically to the clean sequence $\mathbf{x}^0$ and noise sequence $\boldsymbol{\epsilon}$ via the following linear transformations:

$$\mathbf{v} = \boldsymbol{\epsilon} - \mathbf{x}^0 = \frac{\mathbf{x}^{k_\mathcal{T}} - \mathbf{x}^0}{k_\mathcal{T}} = \frac{\boldsymbol{\epsilon} - \mathbf{x}^{k_\mathcal{T}}}{1 - k_\mathcal{T}}, \tag{25}$$

where the division is performed element-wise. Consequently, during inference (Algorithm 3), we can employ networks parameterized to predict either the clean data $\mathbf{x}_\theta^0(\hat{\mathbf{x}}, k_\mathcal{T})$ or the noise $\boldsymbol{\epsilon}_\theta(\hat{\mathbf{x}}, k_\mathcal{T})$. The corresponding update velocity $\mathbf{v}_{\text{pred}}$ is derived as:

$$\mathbf{v}_{\text{pred}} = \frac{\hat{\mathbf{x}} - \mathbf{x}_\theta^0}{k_\mathcal{T}}, \qquad \text{(Data Prediction)} \tag{26}$$

$$\mathbf{v}_{\text{pred}} = \frac{\boldsymbol{\epsilon}_\theta - \hat{\mathbf{x}}}{1 - k_\mathcal{T}}, \qquad \text{(Noise Prediction)} \tag{27}$$

This formulation allows us to flexibly select the prediction target that best suits the modeling requirements.

**Decoupling Temporal and Diffusion Horizons** While Algorithm 1 outlines a simplified inference scheme with a single update per frame, it is often necessary to perform multiple fine-grained denoising steps. In **Algorithm 3**, we relax the discrete streaming index $\tau$ into a continuous domain, subdividing the transition between consecutive audio frames into $S$ Euler sub-steps. Specifically, as the sliding window advances from $\tau$ to $\tau + 1$, we perform $S$ iterative updates using noise schedules interpolated at fractional timestamps $\tau'$. Since each motion token resides within the active window of length $l$ for $l$ streaming steps, this sub-step strategy results in a cumulative total of $S \times l$ denoising iterations per token, corresponding to an effective ODE solver step size of $\delta = (S \cdot l)^{-1}$. Crucially, this design decouples the inference denoising steps from the fixed temporal window used during training. By adjusting $S$, we can flexibly increase the diffusion granularity to achieve high-fidelity generation at inference time, without being constrained by the fixed window size $l$ set during the training phase.

**Temporal Guidance Implementation.** Temporal guidance requires constructing a perturbed history state. Crucially, tokens in the finalized history set $\mathcal{H}$ are already clean ($k \approx 0$). To compute the guidance velocity $\mathbf{v}_{\text{hist}}$, we must explicitly

---

**Algorithm 3** Detailed Streaming Inference with Sub-step Denoising

---

1: **Input:** Audio stream $\mathbf{c}$, Model $\mathbf{v}_\theta$, Max length $T$, Window $l$, **Steps per frame** $S$, Guidance $\omega$.
2: **Initialize:** Latent buffer $\hat{\mathbf{x}} = \emptyset$, Global stream step $\tau \leftarrow 0$.
3: **while** streaming **do**
4:     *// 1. Buffer Maintenance (Shift & Init)*
5:     $\tau_{\text{prev}} \leftarrow \tau$.
6:     $\tau \leftarrow \tau + 1$.
7:     $\hat{\mathbf{x}} \leftarrow [\hat{\mathbf{x}}, \boldsymbol{\epsilon}]$, where $\boldsymbol{\epsilon} \sim \mathcal{N}(\mathbf{0}, \mathbf{I})$.
8:     $\hat{\mathbf{x}} \leftarrow \hat{\mathbf{x}}[-T:]$.                     ▷ Truncate to max length
9:     Define active window $\mathcal{W} \leftarrow \{$indices of last $l$ tokens$\}$.
10:     Define history indices $\mathcal{H} \leftarrow \{$indices before $\mathcal{W}\}$.
11:     *// 2. Multi-step Denoising (Continuous $\tau$ relaxation)*
12:     *// Each frame transition is divided into $S$ sub-steps.*
13:     $\Delta\tau \leftarrow 1/S$.
14:     $\delta \leftarrow \Delta\tau/l$.                    ▷ Effective ODE solver step size
15:     **for** $i = 1$ **to** $S$ **do**
16:         $\tau' \leftarrow \tau_{\text{prev}} + i \cdot \Delta\tau$.         ▷ Fractional stream time $\tau' \in (\tau - 1, \tau]$
17:         *// 2.1 Calculate Noise Levels using Fractional $\tau'$*
18:         $k^{\text{mono}} \leftarrow \Phi_{\text{mono}}(t, \tau', l)$ for all $t$.
19:         $k^{\text{trap}} \leftarrow \Phi_{\text{trap}}(t, \tau', l, l_{\text{ctx}}, l_{\text{hist}})$ for all $t$.
20:         *// 2.2 Construct History-Corrupted State for Guidance*
21:         Sample $\boldsymbol{\epsilon}' \sim \mathcal{N}(\mathbf{0}, \mathbf{I})$.
22:         $\hat{\mathbf{x}}^{\text{trap}} \leftarrow \hat{\mathbf{x}}$.
23:         $\hat{\mathbf{x}}^{\text{trap}}_{\mathcal{H}} \leftarrow \alpha(k^{\text{trap}}_{\mathcal{H}}) \odot \hat{\mathbf{x}}_{\mathcal{H}} + \sigma(k^{\text{trap}}_{\mathcal{H}}) \odot \boldsymbol{\epsilon}'_{\mathcal{H}}$.     ▷ Re-noise clean history
24:         $\hat{\mathbf{x}}^{\text{trap}}_{\mathcal{W}} \leftarrow \hat{\mathbf{x}}_{\mathcal{W}}$.                ▷ Active window remains as is
25:         *// 2.3 Predict Velocity with Temporal Guidance*
26:         $\mathbf{v}_{\text{cond}} \leftarrow \mathbf{v}_\theta(\hat{\mathbf{x}}^{\text{trap}}, k^{\text{trap}}, \mathbf{c})$
27:         $\mathbf{v}_{\text{hist}} \leftarrow \mathbf{v}_\theta(\hat{\mathbf{x}}, k^{\text{mono}}, \varnothing)$
28:         $\mathbf{v} \leftarrow \mathbf{v}_{\text{hist}} + \omega \cdot (\mathbf{v}_{\text{cond}} - \mathbf{v}_{\text{hist}})$.
29:         *// 2.4 Parallel Update (Euler Step)*
30:         $\hat{\mathbf{x}}_{\mathcal{W}} \leftarrow \hat{\mathbf{x}}_{\mathcal{W}} - \mathbf{v}_{\mathcal{W}} \cdot \delta$.
31:     **end for**
32:     *// 3. Emission and Decoding*
33:     Emit clean latent token $\hat{\mathbf{z}} \leftarrow \hat{\mathbf{x}}_{\tau - l + 1}$.         ▷ Leftmost token is now clean
34:     Yield $\hat{\mathbf{m}} \leftarrow \mathcal{D}(\hat{\mathbf{z}})$       ▷ Decode back to 272D motion space for streaming recovery.
35: **end while**

---

re-corrupt these tokens to the target level $k^{\text{trap}}$ using the forward diffusion equation. In contrast, tokens in the active window $\mathcal{W}$ are inherently noisy and do not require additional corruption. This ensures the model receives a consistent input distribution where the history is effectively masked by high-frequency noise, forcing reliance on the low-frequency motion priors.

**Implementation Details of Streaming Audio-driven Motion Diffusion**   Our motion VAE, based on the implementation in Wan2.1 (Wang et al., 2025a), compresses motion sequences into a latent space with dimensionality $Dz = 4$ and a temporal downsampling factor $f = 4$. During training, we slice the raw motion sequences into fixed 64-frame windows. We optimize the VAE using the AdamW optimizer with a learning rate of $5 \times 10^{-4}$, which follows a cosine annealing schedule ($T_{\text{max}} = 1000$, $\eta_{\text{min}} = 10^{-6}$). The training runs for 150k steps with a batch size of 2048. Our transformer diffusion backbone is configured with 8 layers, 8 attention heads, a hidden dimension of 1024 and a feed-forward expansion to 2048. The model is trained to predict velocity using AdamW with a learning rate of $2 \times 10^{-4}$ and the same per-step cosine annealing schedule as the VAE. During train and inference, we apply temporal guidance with a condition dropout rate of 0.1 and a guidance scale of 2.5, perform 10 denoising steps, condition on a history context of 60 latent tokens, and set schedule probabilities ($\pi_{\text{rand}}, \pi_{\text{mono}}, \pi_{\text{trap}}$) to $(0.25, 0.5, 0.25)$. We train for 300k steps with a batch size of 8. All experiments were conducted on NVIDIA RTX 4090 GPUs.

## A.4. Details of Causal Music Processing

**Causal Training and Streaming Cache.** We train the VQ-PAE with a combined objective that encourages accurate audio reconstruction, effective quantization, and temporally smooth periodic features, using a causal encoder–decoder architecture. During online inference, we maintain a FIFO buffer $\mathcal{B}_t$. Let $i$ be the current motion frame index. We align the extraction timestamp by

$$t_{\text{center}} = \min(i + 1, \ T_{\text{total}} - 1), \tag{28}$$

and handle cold-start by zero-padding the unavailable prefix:

$$\mathcal{B}_t = [\underbrace{0, \ldots, 0}_{\text{unobserved}}, \ \underbrace{\mathbf{x}_1, \ldots, \mathbf{x}_{\ell_t}}_{\text{recorded}}], \tag{29}$$

so that the system can emit causal music features immediately without waiting for the full window.

Traditional music-to-motion methods rely on hand-crafted features from librosa (McFee et al., 2015) (e.g., MFCC, chroma, onset envelope), which require manual feature engineering and lack end-to-end optimization for motion generation. Moreover, these features are typically extracted offline on full sequences, making them incompatible with real-time streaming scenarios. To address these limitations, we introduce a Vector Quantized Phase Autoencoder (VQ-PAE) that learns music representations optimized for motion generation while supporting causal, frame-by-frame inference through a music cache manager.

**VQ-PAE Training Objective.** The training objective of our Vector Quantized Phase Autoencoder combines reconstruction, quantization, and phase consistency losses:

$$\mathcal{L} = \mathcal{L}_{\text{recon}} + \lambda_1 \mathcal{L}_{\text{commit}} + \lambda_2 \mathcal{L}_{\text{codebook}} + \lambda_3 \mathcal{L}_{\text{phase}}. \tag{30}$$

The reconstruction loss measures standard L1 distance between input and reconstructed audio: $\mathcal{L}_{\text{recon}} = \|\mathbf{x} - D_\theta(\mathbf{f}_{\text{comb}})\|_1$, where $D_\theta$ is the decoder and $\mathbf{f}_{\text{comb}}$ is the combined latent representation from both VQ and PAE pathways. Following standard VQ-VAE training (Van Den Oord et al., 2017), we employ commitment loss $\mathcal{L}_{\text{commit}} = \|\text{sg}[\mathbf{f}^{\text{causal}}] - \mathbf{f}_{\text{vq}}\|_2^2$ to encourage encoder outputs to stay close to quantized values, and codebook loss $\mathcal{L}_{\text{codebook}} = \|\mathbf{f}^{\text{causal}} - \text{sg}[\mathbf{f}_{\text{vq}}]\|_2^2$ to update the codebook embeddings, where $\text{sg}[\cdot]$ denotes stop-gradient operation. The phase matching loss $\mathcal{L}_{\text{phase}} = \|\mathbf{f}_{\text{recon}} - \text{Phase}(\phi, f, a, b)\|_2^2$ ensures temporal smoothness in the reconstructed periodic signal, where $\text{Phase}(\cdot)$ reconstructs the signal from phase parameters representing phase $\phi$, frequency $f$, amplitude $a$, and offset $b$.

**Architecture Specifications.** The encoder uses 1D convolutions with residual connections and dilation rates [9,3,1], progressively downsampling the input while expanding the receptive field. The decoder mirrors this structure with dilation rates [1,3,9] for upsampling. We use Snake activation functions (Ziyin et al., 2020) throughout the network for better gradient flow and periodic signal modeling. The residual vector quantizer employs $N = 9$ codebooks, each containing $K = 128$ codes of dimension $D = 8$, providing a codebook capacity of $128^9 \approx 9.2 \times 10^{18}$ unique representations.

**Training Configuration.** We train the VQ-PAE on 6-second audio windows sampled at 16kHz. The loss weights are set to $\lambda_1 = 0.25$, $\lambda_2 = 1.0$, and $\lambda_3 = 0.1$. We use the Adam optimizer with learning rate $10^{-4}$ and batch size 32. Training converges after approximately 200 epochs on the AIST++ dataset.

**Frame-Rate Synchronization for Real-Time Streaming.** A critical challenge in real-time music-to-motion generation is maintaining temporal coherence while processing audio in a causal, frame-by-frame manner. Our music cache manager addresses this through a sliding window mechanism that provides sufficient temporal context (6 seconds) for robust feature extraction while enabling immediate response with zero algorithmic latency. The cache manager bridges the gap between audio sample rate (16kHz) and motion frame rate (30 FPS): each motion frame corresponds to $T_f = 533$ audio samples, computed as $T_f = f_{\text{audio}}/f_{\text{motion}} = 16000/30$. Since VQ-PAE processes the entire 6-second window and outputs latent features at the original temporal resolution (96000 dimensions), we employ adaptive average pooling $\mathbf{c}_{\text{pooled}} = \text{AdaptiveAvgPool1d}(\mathbf{f}_{\text{vq/pae}}, 180)$ to downsample to 180 frames, matching our target motion frame rate. This pooling operation preserves temporal coherence while ensuring feature-motion alignment, with each output frame aggregating approximately 533 consecutive input features.

---

**Algorithm 4** 272-D Motion Process from SMPL Sequence (Face-$z^+$ Canonicalization)

---

1: **Input:** SMPL sequence $\{\boldsymbol{\theta}_t, \mathbf{t}_t, \boldsymbol{\beta}_t\}_{t=1}^T$, where $\boldsymbol{\theta}_t \in \mathbb{R}^{22 \times 3}$ (axis-angle, root first), $\mathbf{t}_t \in \mathbb{R}^3$, $\boldsymbol{\beta}_t \in \mathbb{R}^{10}$.
2: **Options:** $\mathtt{mirror} \in \{0, 1\}$.
3: **Output:** 272-D motion features $\mathbf{X} \in \mathbb{R}^{T \times (8+12J)}$ with $J = 22$.
4: **Initialize:** SMPL body model $\mathtt{BM} \leftarrow \mathrm{GetBodyModel}(\mathtt{batch} = T)$.
5: *// 1) Canonicalize sequence to face $z^+$ using first-frame heading*
6: $\mathbf{q}_0 \leftarrow \mathrm{ExpMapToQuat}(\boldsymbol{\theta}_1[0])$       ▷ root orient at $t = 1$
7: $(\mathbf{q}_{\mathrm{inv}}, \mathbf{a}) \leftarrow \mathrm{CalcHeadingQuatInv}(\mathbf{q}_0)$.
8: $\mathbf{q}_\Delta \leftarrow \mathrm{AxisAngleToQuat}(\mathbf{q}_{\mathrm{inv}} \cdot \mathbf{a})$.
9: **for** $t = 1$ **to** $T$ **do**
10:   $\boldsymbol{\theta}'_t[0] \leftarrow \mathrm{QuatMul}(\mathbf{q}_\Delta, \mathrm{ExpMapToQuat}(\boldsymbol{\theta}_t[0]))$     ▷ update root
11:   $\boldsymbol{\theta}'_t[1:] \leftarrow \boldsymbol{\theta}_t[1:]$      ▷ keep local body pose
12:   $\mathbf{t}'_t \leftarrow \mathrm{QuatRot}(\mathbf{q}_\Delta, \mathbf{t}_t)$
13:   $\boldsymbol{\beta}'_t \leftarrow \boldsymbol{\beta}_t$
14: **end for**
15: *// 2) Optional mirroring (left-right swap)*
16: **if** $\mathtt{mirror} = 1$ **then**
17:   $(\boldsymbol{\theta}'_{1:T}, \mathbf{t}'_{1:T}) \leftarrow \mathrm{SwapLeftRightSMPL}(\boldsymbol{\theta}'_{1:T}, \mathbf{t}'_{1:T})$
18: **end if**
19: *// 3) Forward SMPL to obtain global joints (first 22 joints)*
20: $\mathbf{J}_{1:T} \leftarrow \mathtt{BM}(\boldsymbol{\theta}'_{1:T}, \boldsymbol{\beta}'_{1:T}, \mathbf{t}'_{1:T}).\mathtt{Jtr}$
21: $\mathbf{J}_{1:T} \leftarrow \mathbf{J}_{1:T}[:, 1:J]$       ▷ $\mathbf{J} \in \mathbb{R}^{T \times J \times 3}$
22: *// 4) Extract 272-D features (root-velocity + heading-delta + no-heading pos/vel + rot6d)*
23: $\mathbf{X} \leftarrow \mathrm{Extract272}(\mathbf{J}_{1:T}, \boldsymbol{\theta}'_{1:T})$
24: **return** $\mathbf{X}$.

---

**Memory Efficiency and Performance.** The memory footprint of our caching system is highly efficient: the current 6-second audio buffer occupies 96KB (96000 samples × float32), extracted feature vector per frame requires 11.52KB (180 frames × 16 channels × float32), and the total VQ-PAE model contains approximately 12M parameters requiring 48MB. This lightweight design enables deployment on consumer-grade GPUs with as little as 2GB VRAM. On an NVIDIA RTX 3090, we achieve inference latency below 10ms per frame, satisfying real-time constraints with a comfortable margin (30 FPS requires 33ms per frame).

**Audio Preprocessing Pipeline.** To ensure robustness across diverse input conditions, the cache manager incorporates several preprocessing steps executed transparently during streaming. Audio signals are automatically resampled to 16kHz using polyphase filtering if the input sample rate differs, ensuring consistent temporal resolution for VQ-PAE processing. For multi-channel inputs (stereo or surround), we convert to mono by averaging across channels: $x_{\mathrm{mono}}[n] = \frac{1}{C} \sum_{c=1}^C x_c[n]$ where $C$ is the number of channels. When no input is detected (RMS energy below threshold $\tau = 0.01$), the system optionally pads with silence rather than zero-padding, maintaining a consistent audio profile. Input audio is peak-normalized to [-1, 1] range to match training distribution.

**System Integration.** Together, these design choices—the sliding window strategy, zero-padding cold start, frame-rate synchronization, and adaptive pooling—enable our causal music processing pipeline to generate motion in real-time while maintaining temporal coherence. The hybrid VQ-PAE representation provides both the discrete rhythmic precision (via vector quantization) and continuous phase dynamics (via phase parameters) necessary for realistic motion synthesis. Our implementation is based on DeepPhase4Audio (HX-Tfd, 2024) with modifications for real-time streaming inference, including the addition of the music cache manager and optimized CUDA kernels for adaptive pooling operations.

### A.5. Motion Representation and Stream Recovery

Prior dance generation methods (Tseng et al., 2023; Li et al., 2024b) often model motion in global, non-canonical coordinates. In autoregressive (AR) rollouts, global-frame error accumulation induces a distribution shift in the conditioning history, degrading long-horizon stability (e.g., drift and jitter). We therefore adopt a *canonicalized incremental* representation that

---

**Algorithm 5** Extract272: Build 272-D Representation from Joints and SMPL Pose

---

1: **Input:** Global joints $\mathbf{J} \in \mathbb{R}^{T \times J \times 3}$, pose axis-angle $\boldsymbol{\theta} \in \mathbb{R}^{T \times J \times 3}$.
2: **Output:** $\mathbf{X} \in \mathbb{R}^{T \times (8+12J)}$.
3: **Constants:** root index $r \leftarrow 0$.
4: $\mathbf{R} \leftarrow \text{QuatToMat}(\text{ExpMapToQuat}(\boldsymbol{\theta})) \in \mathbb{R}^{T \times J \times 3 \times 3}$.
5: *// 1) Put on floor and normalize initial root to origin*
6: $\mathbf{o} \leftarrow \mathbf{J}_{1,r}; \quad y_{\min} \leftarrow \min(\mathbf{J}_{:,:,y}); \quad o_y \leftarrow y_{\min}$.
7: $\mathbf{J} \leftarrow \mathbf{J} - \mathbf{o}$.
8: $\Delta \mathbf{J}_t^{\text{root}} \leftarrow \mathbf{J}_{t+1,r} - \mathbf{J}_{t,r}$ for $t = 1..T - 1$.
9: *// 2) Localize positions on XZ origin (remove per-frame root XZ)*
10: $\mathbf{J}_{t,:,x} \leftarrow \mathbf{J}_{t,:,x} - \mathbf{J}_{t,r,x}; \quad \mathbf{J}_{t,:,z} \leftarrow \mathbf{J}_{t,:,z} - \mathbf{J}_{t,r,z}$.
11: *// 3) Compute global heading from root rotation*
12: $h_t \leftarrow -\text{atan2}(\mathbf{R}_{t,r}[0,2], \mathbf{R}_{t,r}[2,2])$.
13: $\mathbf{H}_t \leftarrow \text{RotYaw}(h_t) \in \mathbb{R}^{3 \times 3}$.
14: $\Delta h_t \leftarrow h_{t+1} - h_t, \quad \Delta \mathbf{H}_t \leftarrow \text{RotYaw}(\Delta h_t)$ for $t = 1..T - 1$.
15: *// 4) Remove heading for positions and velocities*
16: $\mathbf{P}_{t,j}^{\text{nh}} \leftarrow \mathbf{H}_t \mathbf{J}_{t,j}$ for all $t, j$.
17: $\Delta \mathbf{P}_{t,j}^{\text{nh}} \leftarrow \mathbf{P}_{t+1,j}^{\text{nh}} - \mathbf{P}_{t,j}^{\text{nh}}$ for $t = 1..T - 1$.
18: $\mathbf{v}_{\text{nh},t}^{\text{root}} \leftarrow (\mathbf{H}_t \Delta \mathbf{J}_t^{\text{root}})_{xz}$ for $t = 1..T - 1$.
19: *// 5) Remove heading for root rotation (store no-heading rot6d)*
20: $\mathbf{R}_{t,r} \leftarrow \mathbf{H}_t \mathbf{R}_{t,r}$.
21: *// 6) Pack feature vector*
22: $\mathbf{X} \leftarrow \mathbf{0} \in \mathbb{R}^{T \times (8+12J)}$.
23: Set identity heading delta at first frame: $\mathbf{X}_{1,2} = 1, \mathbf{X}_{1,6} = 1$.
24: $\mathbf{X}_{t+1,0:2} \leftarrow \mathbf{v}_{\text{nh},t}^{\text{root}}$ for $t = 1..T - 1$.
25: $\mathbf{X}_{t+1,2:8} \leftarrow \text{MatToRot6D}(\Delta \mathbf{H}_t)$ for $t = 1..T - 1$.
26: $\mathbf{X}_{t,8:8+3J-1} \leftarrow \text{vec}(\mathbf{P}_{t,:,:}^{\text{nh}})$ for $t = 1..T$.
27: $\mathbf{X}_{t+1,8+3J:8+6J-1} \leftarrow \text{vec}(\Delta \mathbf{P}_{t,:,:}^{\text{nh}})$ for $t = 1..T - 1$.
28: $\mathbf{X}_{t,8+6J:8+12J-1} \leftarrow \text{vec}(\text{MatToRot6D}(\mathbf{R}_{t,:,:}))$ for $t = 1..T$.
29: **return X**.

---

expresses dynamics in the root frame with velocity-like increments, yielding more stationary AR conditioning.

In addition, previous streaming motion generators (Tevet et al., 2025; Chen et al., 2024b) rely on the 263-D HumanML3D (Guo et al., 2022) feature, which typically supports online recovery of joint positions but not the joint rotations required for real-time retargeting, thus necessitating costly post-processing (Bogo et al., 2016). To remove this bottleneck, we adopt the 272-D motion representation (Xiao et al., 2025), which factorizes global locomotion into (i) a planar root translation, (ii) a yaw-only heading evolution, and (iii) heading-invariant joint geometry and rotations. Concretely, for $J{=}22$ joints, each frame is encoded as $\mathbf{x}^{(t)} \in \mathbb{R}^{8+12J}$ with the layout described in Alg. 4: root velocity in XZ under a *no-heading* frame, heading increment as a 6D rotation, per-joint *no-heading* positions and velocities, and per-joint rotations in 6D.

**SMPL-to-272 Motion Processing.** Given an SMPL sequence $\{\boldsymbol{\theta}_t, \mathbf{t}_t, \boldsymbol{\beta}_t\}_{t=1}^{T}$ (axis-angle pose with root first, translation, and shape), we first canonicalize the sequence to *face $z^+$* by removing the first-frame heading. Specifically, we compute an inverse-heading quaternion from the first-frame root orientation and left-multiply it to all root orientations while rotating the global translations accordingly (shape is unchanged). Optionally, we apply left–right mirroring via an SMPL swap operator. We then forward the SMPL body model to obtain global joint positions (we keep the first $J{=}22$ joints) and extract 272-D features from joints and poses; the full pipeline is summarized in Alg. 4.

**Heading Removal and Feature Packing.** Let $\mathbf{J}_{t,j} \in \mathbb{R}^3$ be the global joint position of joint $j$ at time $t$, and let $\mathbf{R}_{t,j} \in \mathbb{R}^{3 \times 3}$ be the rotation matrix converted from the SMPL axis-angle pose. We place the sequence on the floor and normalize the initial root to the origin by subtracting the first-frame root position while setting its $y$ component to the global minimum height. Next, we localize every frame on the XZ origin by removing the per-frame root XZ displacement. We compute the global heading angle from the root rotation matrix via $h_t = -\text{atan2}(\mathbf{R}_{t,r}[0,2], \mathbf{R}_{t,r}[2,2])$ and construct the

yaw rotation $\mathbf{H}_t = \text{RotYaw}(h_t)$, as well as the heading increment $\Delta\mathbf{H}_t = \text{RotYaw}(h_{t+1} - h_t)$. We then remove heading for positions and velocities:

$$\mathbf{P}_{t,j}^{\text{nh}} = \mathbf{H}_t \mathbf{J}_{t,j}, \tag{31}$$

$$\Delta\mathbf{P}_{t,j}^{\text{nh}} = \mathbf{P}_{t+1,j}^{\text{nh}} - \mathbf{P}_{t,j}^{\text{nh}}, \tag{32}$$

$$\mathbf{v}_{\text{root},t}^{\text{nh}} = \left(\mathbf{H}_t(\mathbf{J}_{t+1,r} - \mathbf{J}_{t,r})\right)_{xz}. \tag{33}$$

Finally, we remove heading for the root rotation by left-multiplying $\mathbf{H}_t$ to the root rotation, and pack the feature vector as

$$\mathbf{x}^{(t)} = \left[\mathbf{v}_{\text{root},t-1}^{\text{nh}}, \; \text{Rot6D}(\Delta\mathbf{H}_{t-1}), \; \text{vec}(\mathbf{P}_{t,:}^{\text{nh}}), \; \text{vec}(\Delta\mathbf{P}_{t-1,:}^{\text{nh}}), \; \text{vec}(\text{Rot6D}(\mathbf{R}_{t,:}))\right],$$

with the first-frame heading increment set to identity in 6D for a well-defined stream start (Alg. 4).

**Streaming Recovery from 272-D Frames.**   To recover world-space joints online, we maintain a minimal state consisting of a heading matrix $\mathbf{H} \in \mathbb{R}^{3\times3}$, an accumulated root translation $\mathbf{r} \in \mathbb{R}^3$ (with $r_y{=}0$), a delayed heading delta $\mathbf{D} \in \mathbb{R}^{3\times3}$, and an optional EMA buffer for smoothing. The per-frame streaming update is summarized in Alg. 6. For each incoming frame $\mathbf{x}^{(t)}$, we parse $\mathbf{v}^{\text{nh}} = (x_0, x_1)$, the heading delta in 6D $x_{2:7}$, and the no-heading joint positions $\mathbf{p}^{\text{nh}} = \text{reshape}(x_{8:8+3J-1}, (J, 3))$, and convert the 6D heading delta into $\mathbf{D}_{\text{curr}}$ via Gram–Schmidt.

A key detail is the *offline-compatible delayed heading update*: we integrate root translation using the previous heading (current $\mathbf{H}$), and only then update heading by applying the *previous* delta $\mathbf{D}$, matching the semantics that velocities at time $t$ are expressed in the $t{-}1$ heading frame (Alg. 6). Concretely, we compute $\mathbf{v} = (v_x, 0, v_z)$, map it to world space by $\mathbf{v}_{\text{world}} = \mathbf{H}^\top \mathbf{v}$, and accumulate $\mathbf{r} \leftarrow \mathbf{r} + \mathbf{v}_{\text{world}}$ with $r_y{=}0$. We then update heading as $\mathbf{H} \leftarrow \mathbf{D}\mathbf{H}$ and recover joints by undoing heading and adding translation:

$$\mathbf{P}_{\text{root}} = \left(\mathbf{H}^\top(\mathbf{p}^{\text{nh}})^\top\right)^\top \in \mathbb{R}^{J\times3}, \tag{34}$$

$$\mathbf{P}_{:,x} \leftarrow \mathbf{P}_{:,x} + r_x, \qquad \mathbf{P}_{:,z} \leftarrow \mathbf{P}_{:,z} + r_z. \tag{35}$$

Finally, we optionally apply EMA smoothing and set $\mathbf{D} \leftarrow \mathbf{D}_{\text{curr}}$ for the next frame (Alg. 6).

When the per-joint rotation 6D block is present, we additionally reshape $x_{8+6J:8+12J-1}$ into $(J, 6)$, convert to rotation matrices, apply the recovered heading to the root rotation, and convert to axis-angle to obtain $\boldsymbol{\xi}^{(t)} \in \mathbb{R}^{3J}$. The emitted root translation is set to $(r_x, P_{0,y}, r_z)$, where $P_{0,y}$ is the recovered root height in the current frame (Alg. 6).

### A.6. Extended Training for Standing-Pose Transition

Most prior music-to-dance systems (Tseng et al., 2023; Li et al., 2024b) are trained and evaluated in an *offline* setting: they aim to fit the distribution of a curated dance dataset and are typically assessed by global metrics or short-horizon reconstructions, without requiring long-horizon autoregressive rollouts or real-time, closed-loop consumption in an interactive system. In contrast, our setting is explicitly *streaming*: the model is queried continuously, and its outputs are directly rendered on an avatar or executed by a downstream controller. This makes the behavior under *missing or weak conditions* a first-order concern.

In practice, classifier-free guidance (CFG) (Ho & Salimans, 2021) exposes a critical failure mode for pure dance-distribution fitting. Even when the music condition is empty (i.e., the unconditional branch), a model trained only on dance data tends to generate *unstructured, restless motion* rather than remaining still. This behavior is a benign artifact in offline sampling, but it is unacceptable in interactive avatar and real-robot demos, where the default action under no signal should be a stable standing pose with small natural sway at most.

**Multi-Dataset Extension with BABEL Standing Segments.**   To enforce a well-defined "no-music" fallback, we extend training with an additional corpus of *standing* motion segments mined from BABEL (Punnakkal et al., 2021). BABEL is a large-scale natural-language annotation layer built on top of AMASS (Mahmood et al., 2019), providing both sequence-level action labels and temporally localized action segments (dense event annotations) for mocap clips. In particular, the dataset includes a "stand" action class that captures stationary standing behaviors across diverse subjects, capture setups, and motion styles. We collect these "stand"-labeled temporal intervals and treat them as a complementary distribution that represents the desired idle behavior.

During training, we mix dance sequences and BABEL standing segments (multi-dataset training) and explicitly associate the standing segments with an *null-music* condition. This teaches the model to map the null-music condition to a stable pose manifold, so that when the music stream is absent (or when guidance steers toward the unconditional prediction), the generator produces a coherent standing posture instead of drifting into random dance-like motions. Empirically, this modification substantially improves the perceived stability and usability of both the online avatar platform and the physics-based deployment pipeline.

### A.7. Metrics Calculation

We compute all music-to-dance metrics by strictly following the evaluation protocol and reference implementation of LODGE (Li et al., 2024b). Given a motion sequence as global joint positions $\mathbf{P} \in \mathbb{R}^{T \times J \times 3}$, we first convert it into a *root-relative* representation by subtracting the root joint position for all non-root joints:

$$\tilde{\mathbf{P}}_{t,0} = \mathbf{P}_{t,0}, \qquad \tilde{\mathbf{P}}_{t,j} = \mathbf{P}_{t,j} - \mathbf{P}_{t,0}, \ \forall j \in \{1, \ldots, J-1\}. \tag{36}$$

This root-relative joint trajectory is used for both kinetic and manual (geometric) feature extraction.

**Kinetic Feature Extraction.** Following LODGE, we compute a kinetic feature vector by concatenating per-joint statistics derived from average velocities/accelerations over time. Let $\Delta t = 1/30$ and a short sliding window size $w = 2$ (as in the code). Denote the (windowed) average velocity and acceleration of joint $j$ at frame $t$ as $\bar{\mathbf{v}}_{t,j}$ and $\bar{\mathbf{a}}_{t,j}$, respectively. We decompose the velocity into horizontal (ground-plane) and vertical components under the $y$-up convention: $\bar{\mathbf{v}}_{t,j}^{\text{hor}} = (\bar{v}_{t,j,x}, \bar{v}_{t,j,z})$ and $\bar{v}_{t,j}^{\text{ver}} = \bar{v}_{t,j,y}$. The three scalar kinetic descriptors for each joint $j$ are then:

$$E_j^{\text{hor}}(\tilde{\mathbf{P}}) = \frac{1}{T-1} \sum_{t=1}^{T-1} \left\| \bar{\mathbf{v}}_{t,j}^{\text{hor}} \right\|_2^2, \tag{37}$$

$$E_j^{\text{ver}}(\tilde{\mathbf{P}}) = \frac{1}{T-1} \sum_{t=1}^{T-1} \left( \bar{v}_{t,j}^{\text{ver}} \right)^2, \tag{38}$$

$$A_j(\tilde{\mathbf{P}}) = \frac{1}{T-1} \sum_{t=1}^{T-1} \left\| \bar{\mathbf{a}}_{t,j} \right\|_2, \tag{39}$$

corresponding to average horizontal kinetic energy, average vertical kinetic energy, and average energy expenditure, respectively. The final kinetic feature is $\phi_k(\tilde{\mathbf{P}}) = \left[ E_j^{\text{hor}}, E_j^{\text{ver}}, A_j \right]_{j=0}^{J-1} \in \mathbb{R}^{3J}$.

**Manual (Geometric) Feature Extraction.** We also compute a geometric/manual feature vector based on a fixed set of rule-based pose and motion predicates over SMPL-style joints (e.g., wrist motion relative to torso, distance-to-plane constraints, joint angle ranges, and fast-motion indicators). Let $\{g_m(\tilde{\mathbf{P}}_t, \tilde{\mathbf{P}}_{t-1})\}_{m=1}^M$ be these per-frame predicate outputs (typically binary indicators or thresholded values). The manual feature aggregates them by temporal averaging:

$$\phi_g(\tilde{\mathbf{P}}) = \frac{1}{T-1} \sum_{t=1}^{T-1} \mathbf{g}(\tilde{\mathbf{P}}_t, \tilde{\mathbf{P}}_{t-1}) \in \mathbb{R}^M, \tag{40}$$

which matches the implementation that iterates over frames, collects predicate values, and averages over time.

**Feature Normalization.** Before computing distributional metrics, we normalize generated features using the ground-truth feature mean and standard deviation:

$$\hat{\phi} = \frac{\phi - \boldsymbol{\mu}_{\text{gt}}}{\boldsymbol{\sigma}_{\text{gt}} + \epsilon}, \qquad \boldsymbol{\mu}_{\text{gt}} = \frac{1}{N} \sum_{i=1}^N \phi_{\text{gt}}^{(i)}, \qquad \boldsymbol{\sigma}_{\text{gt}} = \sqrt{\frac{1}{N} \sum_{i=1}^N \left( \phi_{\text{gt}}^{(i)} - \boldsymbol{\mu}_{\text{gt}} \right)^2}, \tag{41}$$

as in the LODGE-derived codepath.

**FID in Kinetic/Manual Feature Spaces.** Let $\{\hat{\phi}_{\text{gt}}^{(i)}\}_{i=1}^N$ and $\{\hat{\phi}_{\text{gen}}^{(i)}\}_{i=1}^{N'}$ be normalized features (either $\hat{\phi}_k$ or $\hat{\phi}_g$). We compute empirical Gaussian statistics:

$$\boldsymbol{\mu} = \frac{1}{N}\sum_{i=1}^N \hat{\phi}^{(i)}, \qquad \boldsymbol{\Sigma} = \frac{1}{N-1}\sum_{i=1}^N \big(\hat{\phi}^{(i)} - \boldsymbol{\mu}\big)\big(\hat{\phi}^{(i)} - \boldsymbol{\mu}\big)^\top. \tag{42}$$

The Frechét distance (FID) between the generated and ground-truth feature distributions is:

$$\text{FID} = \big\|\boldsymbol{\mu}_{\text{gt}} - \boldsymbol{\mu}_{\text{gen}}\big\|_2^2 + \text{Tr}\Big(\boldsymbol{\Sigma}_{\text{gt}} + \boldsymbol{\Sigma}_{\text{gen}} - 2\big(\boldsymbol{\Sigma}_{\text{gt}}\boldsymbol{\Sigma}_{\text{gen}}\big)^{\frac{1}{2}}\Big), \tag{43}$$

yielding $\text{FID}_k$ and $\text{FID}_g$ for kinetic and manual features, respectively, consistent with the implementation that computes activation statistics and applies Frechét distance.

**Diversity in Kinetic/Manual Feature Spaces.** We compute diversity as the average Euclidean distance between feature vectors of randomly sampled pairs. If the number of sequences exceeds a sampling budget $K$ (default $K=300$), we sample $K$ pairs $\{(i_\ell, j_\ell)\}_{\ell=1}^K$ uniformly without replacement and report:

$$\text{DIV} = \frac{1}{K}\sum_{\ell=1}^K \Big\|\hat{\phi}_{\text{gen}}^{(i_\ell)} - \hat{\phi}_{\text{gen}}^{(j_\ell)}\Big\|_2, \tag{44}$$

otherwise we compute the average distance over all pairs (the "avg_distance" fallback). This gives $\text{DIV}_k$ and $\text{DIV}_g$ under kinetic/manual features, respectively.

**Beat Alignment Score (BAS).** We follow the standard BAS computation. Music beats $\mathcal{B}_m$ are extracted from the waveform via onset strength and beat tracking (with 30 FPS-aligned sampling rate and fixed hop length). Motion beats $\mathcal{B}_d$ are detected as local minima of a smoothed kinetic-speed curve:

$$s_t = \frac{1}{J}\sum_{j=0}^{J-1} \|\mathbf{P}_{t+1,j} - \mathbf{P}_{t,j}\|_2, \qquad \tilde{s}_t = G_\sigma(s_t), \qquad \mathcal{B}_d = \{t :\ t \text{ is a local minimum of } \tilde{s}_t\}, \tag{45}$$

where $G_\sigma(\cdot)$ is a Gaussian smoothing operator. Given $\mathcal{B}_m$ and $\mathcal{B}_d$, BAS is:

$$\text{BAS}(\mathcal{B}_m, \mathcal{B}_d) = \frac{1}{|\mathcal{B}_m|}\sum_{b \in \mathcal{B}_m} \exp\Big(-\frac{\min_{t \in \mathcal{B}_d}(t-b)^2}{2\sigma_b^2}\Big), \qquad \sigma_b^2 = 9, \tag{46}$$

matching the released implementation.

**Foot Skating Ratio (FSR).** To quantify foot sliding artifacts, we compute the foot skating ratio on the generated joint trajectories. Let $y_{t,a}^L$ and $y_{t,\tau}^L$ denote the left ankle/toe heights at frame $t$ (similarly for the right foot). A frame is counted as *foot contact* if both ankle and toe are close to the ground:

$$\mathbb{I}_t^L = \mathbb{I}\big(y_{t,a}^L \le h_g + \delta_a\big) \wedge \mathbb{I}\big(y_{t,\tau}^L \le h_g + \delta_\tau\big), \tag{47}$$

with $h_g=0$, $\delta_a=0.08$ m and $\delta_\tau=0.05$ m. We then compute per-frame tangential displacement in the ground plane for ankle/toe, $\Delta\mathbf{p}_{t,j}^{xz} = (p_{t+1,j,x} - p_{t,j,x},\ p_{t+1,j,z} - p_{t,j,z})$. A contact frame is marked as *sliding* if the mean absolute tangential displacement (averaged over $x/z$) exceeds a small threshold $d_{\text{th}}$:

$$\mathbb{S}_t^L = \mathbb{I}_t^L \wedge \mathbb{I}\bigg(\max_{j \in \{\text{ankle}^L, \text{toe}^L\}} \frac{|\Delta p_{t,j,x}| + |\Delta p_{t,j,z}|}{2} > d_{\text{th}}\bigg), \qquad d_{\text{th}} = 0.01. \tag{48}$$

The left/right skating ratios are $\text{FSR}^L = \frac{\sum_t \mathbb{S}_t^L}{\sum_t \mathbb{I}_t^L + \epsilon}$ and $\text{FSR}^R = \frac{\sum_t \mathbb{S}_t^R}{\sum_t \mathbb{I}_t^R + \epsilon}$, and the final FSR is

$$\text{FSR} = \frac{1}{2}\big(\text{FSR}^L + \text{FSR}^R\big), \tag{49}$$

where lower is better. (In our evaluation pipeline, FSR is accumulated per sequence and averaged over the dataset.)

### A.8. Robot Control System

Given a generated kinematic reference over a short horizon, we (i) retarget it to the robot configuration space (Araujo et al., 2025), and (ii) execute it using a joint-impedance tracking controller as commonly used in physics-based humanoid systems (Liao et al., 2025)

**Robot Motion Retargeting.** Let $\mathbf{p}_t^{\text{tgt}} \in \mathbb{R}^{3n_j}$ denote the target 3D joint positions at time $t$ (e.g., from an SMPL-to-robot regressed skeleton), and $\text{FK}(\mathbf{p}_t, \mathbf{R}_t, \mathbf{q}_t)$ denote the robot joint positions via forward kinematics, parameterized by root translation $\mathbf{p}_t$, root rotation $\mathbf{R}_t \in SO(3)$, and joint angles $\mathbf{q}_t$. A basic per-sequence retargeting objective minimizes joint position error with joint-limit constraints (Araujo et al., 2025).

$$\min_{\{\mathbf{p}_t, \mathbf{R}_t, \mathbf{q}_t\}_{t=1}^T} \frac{1}{T} \sum_{t=1}^T \left\| \mathbf{p}_t^{\text{tgt}} - \text{FK}(\mathbf{p}_t, \mathbf{R}_t, \mathbf{q}_t) \right\|_2^2, \quad \text{s.t. } \mathbf{q}^- \leq \mathbf{q}_t \leq \mathbf{q}^+. \tag{50}$$

For higher-quality retargeting, one may optimize on a set of key bodies with both orientation and position terms. Let $\mathcal{M}$ be the human→robot body mapping, $\mathcal{M}_{ee} \subset \mathcal{M}$ be end-effectors, $\mathbf{R}_i^h$ be the source (human) body orientation, and $(\mathbf{p}_j(\mathbf{q}), \mathbf{R}_j(\mathbf{q}))$ be the robot body pose under FK. A common two-stage IK uses (i) end-effector positions + all mapped orientations, then (ii) all mapped positions + orientations with different weights (Araujo et al., 2025):

$$\min_{\mathbf{q}} \sum_{(i,j) \in \mathcal{M}} w_{R,ij}^{(1)} \left\| \log\left( (\mathbf{R}_i^h)^\top \mathbf{R}_j(\mathbf{q}) \right) \right\|_2^2 + \sum_{(i,j) \in \mathcal{M}_{ee}} w_{p,ij}^{(1)} \left\| \mathbf{p}_i^{\text{tgt}} - \mathbf{p}_j(\mathbf{q}) \right\|_2^2, \text{ s.t. } \mathbf{q}^- \leq \mathbf{q} \leq \mathbf{q}^+, \tag{51}$$

$$\min_{\mathbf{q}} \sum_{(i,j) \in \mathcal{M}} w_{R,ij}^{(2)} \left\| \log\left( (\mathbf{R}_i^h)^\top \mathbf{R}_j(\mathbf{q}) \right) \right\|_2^2 + \sum_{(i,j) \in \mathcal{M}} w_{p,ij}^{(2)} \left\| \mathbf{p}_i^{\text{tgt}} - \mathbf{p}_j(\mathbf{q}) \right\|_2^2, \text{ s.t. } \mathbf{q}^- \leq \mathbf{q} \leq \mathbf{q}^+. \tag{52}$$

**Robot Tracking Controller.** We execute the retargeted reference using a joint-impedance (PD) controller. Let $\mathbf{q}$ and $\dot{\mathbf{q}}$ be the current joint state, and $\mathbf{q}^{\text{des}}$ be the commanded joint setpoint. The commanded torque is

$$\boldsymbol{\tau} = \mathbf{K}_p(\mathbf{q}^{\text{des}} - \mathbf{q}) - \mathbf{K}_d \dot{\mathbf{q}}. \tag{53}$$

Following common practice, stiffness and damping can be parameterized via natural frequency $\omega_n$ and damping ratio $\zeta$,

$$\mathbf{K}_p = \omega_n^2 \mathbf{I}_{\text{ref}}, \qquad \mathbf{K}_d = 2\zeta\omega_n \mathbf{I}_{\text{ref}}, \tag{54}$$

where $\mathbf{I}_{\text{ref}}$ denotes the reflected joint inertia approximation used for gain scaling (Liao et al., 2025).

We define the single-step observation as the concatenation of: (i) reference phase features (reference joint positions/velocities), (ii) anchor-body pose tracking error, and (iii) proprioceptive features including root twist, joint positions/velocities, and previous action:

$$\mathbf{o}_t = \Big[ \underbrace{\mathbf{q}_t^m, \dot{\mathbf{q}}_t^m}_{\text{reference phase}}, \underbrace{\mathbf{e}_t^{\text{anchor}}}_{\text{anchor error}}, \underbrace{\mathbf{v}_t^{\text{root}}, \mathbf{q}_t, \dot{\mathbf{q}}_t, \mathbf{a}_{t-1}}_{\text{proprioception}} \Big], \tag{55}$$

where $\mathbf{e}_t^{\text{anchor}}$ typically includes 3D position error and a rotation-error representation (e.g., first two columns of a relative rotation matrix).

The policy outputs a normalized action $\mathbf{a}_t \in [-1, 1]^{n_q}$ which is mapped to joint setpoints. A widely used mapping scales by torque limits and stiffness so that the setpoint magnitude corresponds to feasible torque ranges:

$$\mathbf{q}_t^{\text{des}} = \mathbf{q}^{\text{nom}} + \mathbf{a}_t \odot \left( \boldsymbol{\tau}^{\text{max}} \oslash \text{diag}(\mathbf{K}_p) \right), \tag{56}$$

where $\mathbf{q}^{\text{nom}}$ is a nominal joint configuration, $\boldsymbol{\tau}^{\text{max}}$ is the per-joint torque limit, $\odot$ denotes element-wise product, and $\oslash$ denotes element-wise division. The resulting $\mathbf{q}_t^{\text{des}}$ is then tracked by the PD controller in Eq. (53).

## B. Additional Experiments

### B.1. Streaming-Native Baseline Comparisons

Most existing music-to-dance methods are designed for offline generation with access to full audio context. To further evaluate DiscoForcing under a deployment-faithful protocol, we additionally compare against streaming-native causal baselines

*Table 5.* Comparison with streaming-native causal baselines on AIST++ under the same streaming protocol. A higher or lower value is better for ↑ or ↓, and → means the value closer to ground truth is better.

| Method | $\text{FID}_k\downarrow$ | $\text{FID}_g\downarrow$ | FSR↓ | $\text{Div}_k\rightarrow$ | $\text{Div}_g\rightarrow$ | BAS↑ |
|---|---|---|---|---|---|---|
| GT | – | – | 0.007 | 8.19 | 7.45 | 0.237 |
| CLoSD (Tevet et al., 2025) | 27.59 | 14.68 | **0.050** | 7.02 | 6.51 | 0.220 |
| DART (Zhao et al., 2025) | 39.12 | 19.34 | 0.062 | **8.76** | **7.12** | 0.231 |
| MotionStreamer (Xiao et al., 2025) | 29.84 | 20.55 | 0.071 | 6.20 | 4.79 | 0.239 |
| **DiscoForcing** | **18.87** | **11.57** | 0.059 | 6.76 | 6.31 | **0.244** |

*Table 6.* Blinded pairwise human preference study under the streaming setting. We report the percentage of trials in which DiscoForcing is preferred over the compared baseline.

| Comparison | Motion Stability | Beat Alignment | Transition Responsiveness | Overall Preference |
|---|---|---|---|---|
| Ours vs. CLoSD | 66% | 71% | 67% | 73% |
| Ours vs. DARTControl | 68% | 66% | 70% | 74% |
| Ours vs. MotionStreamer | 62% | 69% | 65% | 70% |

adapted to the same audio-driven setting: CLoSD (Tevet et al., 2025), DART (Zhao et al., 2025), and MotionStreamer (Xiao et al., 2025). For fairness, all baselines are evaluated with causal audio only, the same VQ-PAE music encoder, the same 272-D motion representation, and matched 30 Hz streaming constraints. The baselines differ only in the sequence generator and streaming inference strategy.

Table 5 reports the results on AIST++. DiscoForcing achieves the strongest motion fidelity and beat alignment among the streaming baselines, with the best $\text{FID}_k$, $\text{FID}_g$, and BAS. While some baselines obtain competitive foot-skating or diversity scores, DiscoForcing provides the best overall trade-off between motion quality, rhythm synchronization, and real-time streaming deployability.

## B.2. Human Preference Study

Because interactive character control ultimately depends on perceived motion quality, we further conduct a blinded pairwise human study under the same causal streaming setting. We compare DiscoForcing against the three streaming-native baselines in Table 5. Each trial uses the same input audio for both methods, rendered with the same avatar visualization pipeline. The method order is randomized and method names are hidden. The test clips cover diverse music genres and include concatenated excerpts with at least one music transition. We collect responses from 100 participants, who are asked to choose the better result along four axes: motion stability, beat alignment, transition responsiveness, and overall preference.

As shown in Table 6, DiscoForcing is preferred over all compared baselines across all perceptual criteria. These results complement the automatic metrics and indicate that the quantitative improvements translate to perceptible gains under streaming deployment.

## B.3. Physics-Based Humanoid Tracking Evaluation

To complement the qualitative humanoid demonstrations, we provide a quantitative evaluation of the downstream physics-based tracking controller. We evaluate the retargeted reference motions in simulation using the same whole-body control stack described in Appendix A.5. A rollout is counted as successful if the humanoid remains balanced and completes the reference sequence without falling. We report success rate, mean per-joint position error (MPJPE), acceleration tracking error $E_{\text{acc}}$, and velocity tracking error $E_{\text{vel}}$.

As shown in Table 7, the controller tracks DiscoForcing-generated references with high success rate on both train and test splits. This indicates that the generated motions are not only visually plausible for avatar playback, but can also be converted into executable humanoid references with stable closed-loop tracking.

*Table 7.* Quantitative physics-based humanoid tracking evaluation. A higher or lower value is better for ↑ or ↓.

| Method | Succ.↑ | MPJPE↓ | $E_{\text{acc}}$ ↓ | $E_{\text{vel}}$ ↓ |
|---|---|---|---|---|
| Tracking Controller - Train | 94.1% | 48.72 | 8.19 | 7.28 |
| Tracking Controller - Test | 85.3% | 59.91 | 9.56 | 8.71 |

*Table 8.* Latency Table for Interactive System (ms / frame)

| **Music Latency** | | | |
|---|---|---|---|
| Librosa | 9.60 | VQ-PAE | 8.30 |
| **Motion Latency** | | | |
| Motion Decoder | 1.47 | Diffusion Sampling | 24.79 |
| **Application Latency** | | | |
| Unity Render | 0.22 | Online Retargeting | 11.27 |
| Online Interpolation | 5.25 | Tracking Controller | 1.84 |

## B.4. System Latency Analysis

We analyze both module-level runtime and end-to-end streaming latency of the interactive system. All latency numbers are measured on a single NVIDIA GeForce RTX 4090 GPU. Table 8 reports the average per-frame runtime of each system component. The throughput-critical generation path is dominated by diffusion sampling and motion decoding, where diffusion sampling takes 24.79 ms/frame and the motion decoder takes 1.47 ms/frame, resulting in 26.26 ms/frame for motion generation. The learned VQ-PAE music encoder takes 8.30 ms/frame, slightly faster than the Librosa baseline.

Importantly, our system is not executed as a fully serial per-frame pipeline. Instead, audio encoding, motion generation and downstream application asynchronously through ROS2 and communicate continuously. Therefore, after pipeline fill, the steady-state throughput is determined by the slowest asynchronous stage by the slowest stage, i.e., motion generation at 26.26 ms/frame, which is within the 33 ms/frame budget for 30 FPS streaming. The measured first-frame end-to-end latency is 34.78 ms for avatar visualization and 52.92 ms for humanoid execution, corresponding to approximately one frame and less than two frames, respectively.

## B.5. Ablations on Causal Music Encoding

To validate our design choices for the causal music encoding module, we conduct ablation studies comparing different audio representation learning architectures on the AIST++ (Li et al., 2021) dataset. We compare four music encoding architectures: **1) PAE-Vanilla**, the vanilla Phase Autoencoder (Starke et al., 2022) adapted for audio signals, which employs a shallow encoder with two 1D convolution layers to extract phase parameters (phase $\phi$, frequency $f$, amplitude $a$, offset $b$) through FFT analysis and reconstructs the signal via sinusoidal basis functions; **2) PAE-Deep**, an extended version with deeper convolutional layers (4 layers with intermediate channels of 16 and 32) to increase model capacity for capturing complex audio patterns; **3) PAE-Wave**, which incorporates WaveNet-style dilated convolutions (Van Den Oord et al., 2016) with exponentially increasing dilation rates and gated activation units, designed to capture long-range temporal dependencies in audio signals; and **4) VQ-PAE (Ours)**, our proposed Vector Quantized Phase Autoencoder that combines residual vector quantization (Zeghidour et al., 2021) with phase decomposition through a dual-pathway design: a VQ branch for discrete rhythmic pattern encoding and a phase branch for continuous periodic dynamics.

**Audio Reconstruction Quality.** Table 9 presents the audio reconstruction metrics on the AIST++ dataset. We evaluate using Mean Squared Error (MSE), STFT Spectral Convergence (SC), STFT Magnitude Loss (Mag), and Mel Spectrogram Loss (Mel).

The results demonstrate that VQ-PAE significantly outperforms all baseline architectures. While PAE-Deep and PAE-Wave provide marginal improvements in MSE over the vanilla PAE, they exhibit worse spectral reconstruction quality, indicating that simply increasing model capacity does not address the fundamental limitations of sinusoidal basis reconstruction. VQ-PAE achieves substantial improvements across all metrics, reducing MSE by 56% and STFT Spectral Convergence by

*Table 9.* Audio reconstruction quality comparison of different music encoders on AIST++. Lower is better for all metrics.

| Method | Latent Dim | MSE↓ | SC↓ | Mag↓ | Mel↓ |
|---|---|---|---|---|---|
| PAE-Vanilla | 8 | 0.032 | 1.482 | 0.818 | 2.332 |
| PAE-Deep | 32 | 0.030 | 1.556 | 1.716 | 3.302 |
| PAE-Wave | 16 | 0.040 | 2.405 | 3.371 | 5.816 |
| VQ-PAE (Ours) | 8 | **0.014** | **0.063** | **0.495** | **0.390** |

*Table 10.* Motion reconstruction quality comparison of motion encoders on AIST++.

| Input Dim | Latent Dim | $\text{FID}_k$↓ | $\text{FID}_g$↓ | $\text{Div}_k$↑ | $\text{Div}_g$↑ | MPJPE ↓ | PAMPJPE ↓ | ACCL ↓ |
|---|---|---|---|---|---|---|---|---|
| 263 | 4 | 3.420 | 1.315 | 8.7597 | 7.015 | 0.0720 | 0.0163 | 0.0221 |
| 272 | 8 | 10.607 | 1.976 | 9.177 | 7.016 | 0.1554 | 0.0416 | 0.0223 |
| 272 | 4 | 7.993 | 3.447 | 6.789 | 8.472 | 0.1675 | 0.0610 | 0.0239 |

96% compared to PAE-Vanilla.

**Qualitative Results.** PAE-Vanilla produces waveforms with significant phase misalignment and amplitude distortion, failing to capture the fine-grained temporal structure of music signals. Despite increasing network capacity, PAE-Deep shows only marginal improvement, suggesting that the bottleneck lies not in model capacity but in the representational framework. PAE-Wave, while leveraging dilated convolutions for larger receptive fields, still struggles with polyphonic audio due to the inherent limitations of sinusoidal basis reconstruction. In contrast, VQ-PAE produces reconstructions that are virtually indistinguishable from the original waveform, preserving both the envelope dynamics and spectral details.

**Analysis.** The superior performance of VQ-PAE can be attributed to two key design choices. First, residual vector quantization creates a discrete bottleneck that forces the encoder to learn compact, semantically meaningful representations of rhythmic patterns. With 9 codebooks of 128 codes each, VQ-PAE has sufficient capacity to represent the full spectrum of musical patterns while avoiding the mode collapse common in continuous latent spaces. Second, the dual-pathway architecture separates the VQ branch (for discrete rhythm encoding) from the phase branch (for continuous periodic dynamics), allowing our architecture to simultaneously capture both the sharp transients of beats and the smooth evolution of melodic content. In contrast, vanilla PAE variants struggle because the sinusoidal basis functions have limited expressiveness for complex, polyphonic audio signals—increasing network depth (PAE-Deep) or using dilated convolutions (PAE-Wave) provides only incremental improvements, as the fundamental limitation lies in the reconstruction formulation rather than encoder capacity.

### B.6. Ablations on Motion Primitive Learning

Table 10 compares different VAE configurations. While the 263-dim input or 8-dim latent space yield marginally better reconstruction, they are unsuitable for our specific latency-bounded streaming framework. The 263-dim representation lacks joint rotations, necessitating costly Inverse Kinematics (IK) for execution, while 8-dim latents impose excessive computational overhead on the diffusion backbone. Therefore, we select the 272-dim input with a 4-dim latent space, which enables efficient Forward Kinematics and maintains high inference throughput with minimal quality loss.

### B.7. Additional ablations on Model Design and Inference Settings

We investigate the sensitivity of DiscoForcing to temporal guidance weight ($\omega$) and scheduling parameters ($l, l_{ctx}, l_{hist}$). Additionally, we compare different network parameterizations: velocity ($v$), noise ($\epsilon$), and data ($\mathbf{x}^0$) prediction. As shown in Table 11, we find that $v$-prediction yields superior performance compared to other parameterizations. We also observe that a balanced guidance scale is required to achieve optimal fidelity, as extreme values tend to degrade the results. Regarding the temporal settings, we observe that increasing the denoising window $l$ or context lengths also produces competitive generation quality but at cost of higher response latency. Consequently, our chosen configuration represents an optimal balance, achieving state-of-the-art fidelity while maintaining the low latency required for real-time applications.

*Table 11.* Ablation study of the temporal scheduling hyperparameters on AIST++.

| $\omega$ | $l$ | $l_{ctx}$ | $l_{hist}$ | $v/\epsilon/x^0$ | $\text{FID}_k \downarrow$ | $\text{FID}_g \downarrow$ | $\text{FSR}\downarrow$ | $\text{DIV}_k \rightarrow$ | $\text{DIV}_g \rightarrow$ | $\text{BAS}\uparrow$ |
|---|---|---|---|---|---|---|---|---|---|---|
| GROUND TRUTH | | | | | - | - | 0.007 | 8.19 | 7.45 | 0.237 |
| 2.5 | 5 | 5 | 5 | $v$ | 21.10 | 12.24 | 0.076 | 6.53 | 5.75 | 0.245 |
| 2.5 | 10 | 5 | 10 | $v$ | 18.06 | 11.97 | 0.045 | 7.05 | 6.81 | 0.238 |
| 2.5 | 10 | 10 | 10 | $v$ | 23.62 | 9.06 | 0.105 | 6.02 | 7.17 | 0.241 |
| 1.5 | 5 | 1 | 5 | $v$ | 30.10 | 12.80 | 0.086 | 7.06 | 6.07 | 0.247 |
| 5.0 | 5 | 1 | 5 | $v$ | 24.18 | 7.83 | 0.058 | 6.50 | 5.89 | 0.238 |
| 2.5 | 5 | 1 | 5 | $\epsilon$ | 30.65 | 10.86 | 0.125 | 6.78 | 5.87 | 0.240 |
| 2.5 | 5 | 1 | 5 | $x^0$ | 25.96 | 14.47 | 0.066 | 5.83 | 6.17 | 0.250 |
| 2.5 | 5 | 1 | 5 | $v$ | 18.87 | 11.57 | 0.059 | 6.76 | 6.31 | 0.244 |

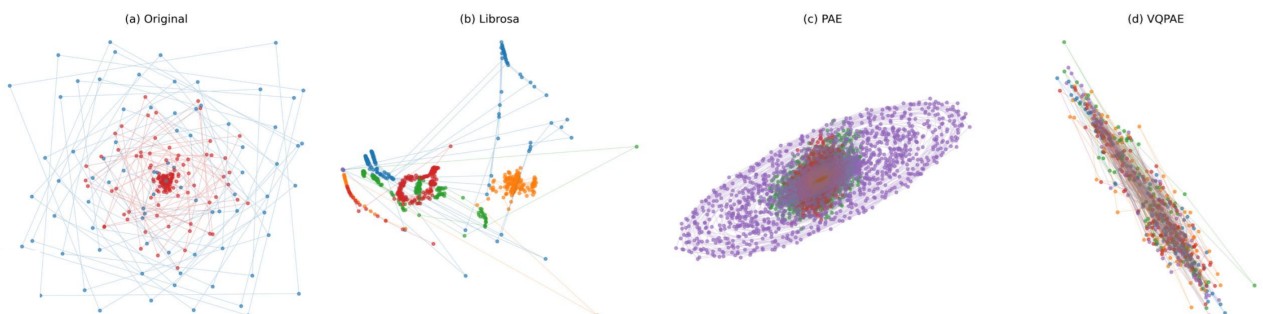

*Figure 5.* Music latent space visualization across different audio representations. Manifold visualizations of five audio clips using four feature extraction methods: (a) Original waveform, (b) Librosa, (c) PAE, and (d) VQPAE. Each color denotes a different audio clip. Raw waveform and Librosa features form scattered or weakly clustered point clouds. In contrast, PAE learns smooth, well-separated spiral phase manifolds, while VQPAE produces more compact and dense circular structures, revealing clear periodic geometry not captured by conventional audio representations.

## C. Additional Results

### C.1. Music Latent Space Visualization

To provide an intuitive comparison of the characteristics of different audio representation methods, we select five audio clips and perform manifold visualization analysis. All audio signals are resampled to 16 kHz. The visualization consists of four subplots, each corresponding to one feature extraction method.

**Original waveform (Original).** We segment each audio signal into frames with a fixed frame length of 2048 samples and a hop size of 512 samples. Each frame is directly treated as a 2048-dimensional feature vector. We then apply principal component analysis (PCA) jointly over all frames from all audio clips and project the features to 2D for visualization. Since raw waveforms lack structured representations, the projected points form a scattered cloud where samples from different clips are heavily mixed, making them difficult to distinguish.

**Hand-crafted acoustic features (Librosa).** We extract a set of classic acoustic features using the Librosa toolkit (McFee et al., 2015), including 12-dimensional chroma features, 7-dimensional spectral contrast, spectral centroid, spectral bandwidth, spectral roll-off, zero-crossing rate, and 6-dimensional tonnetz features. Concatenating these components yields a 30-dimensional feature vector, which is further reduced to 2D using PCA. These features provide a moderate degree of discriminability: frames from the same clip tend to cluster, but the overall distribution remains irregular and does not clearly reveal the periodic structure of the underlying audio signals.

**Periodic autoencoder (PAE).** For each clip, we take the first 2 seconds (32,000 samples) as a single time window and feed it into a pretrained PAE model. The encoder outputs a 16-channel latent representation, where channels are paired as eight $(\sin, \cos)$ component pairs to represent phase information at different frequency components, following prior phase-based

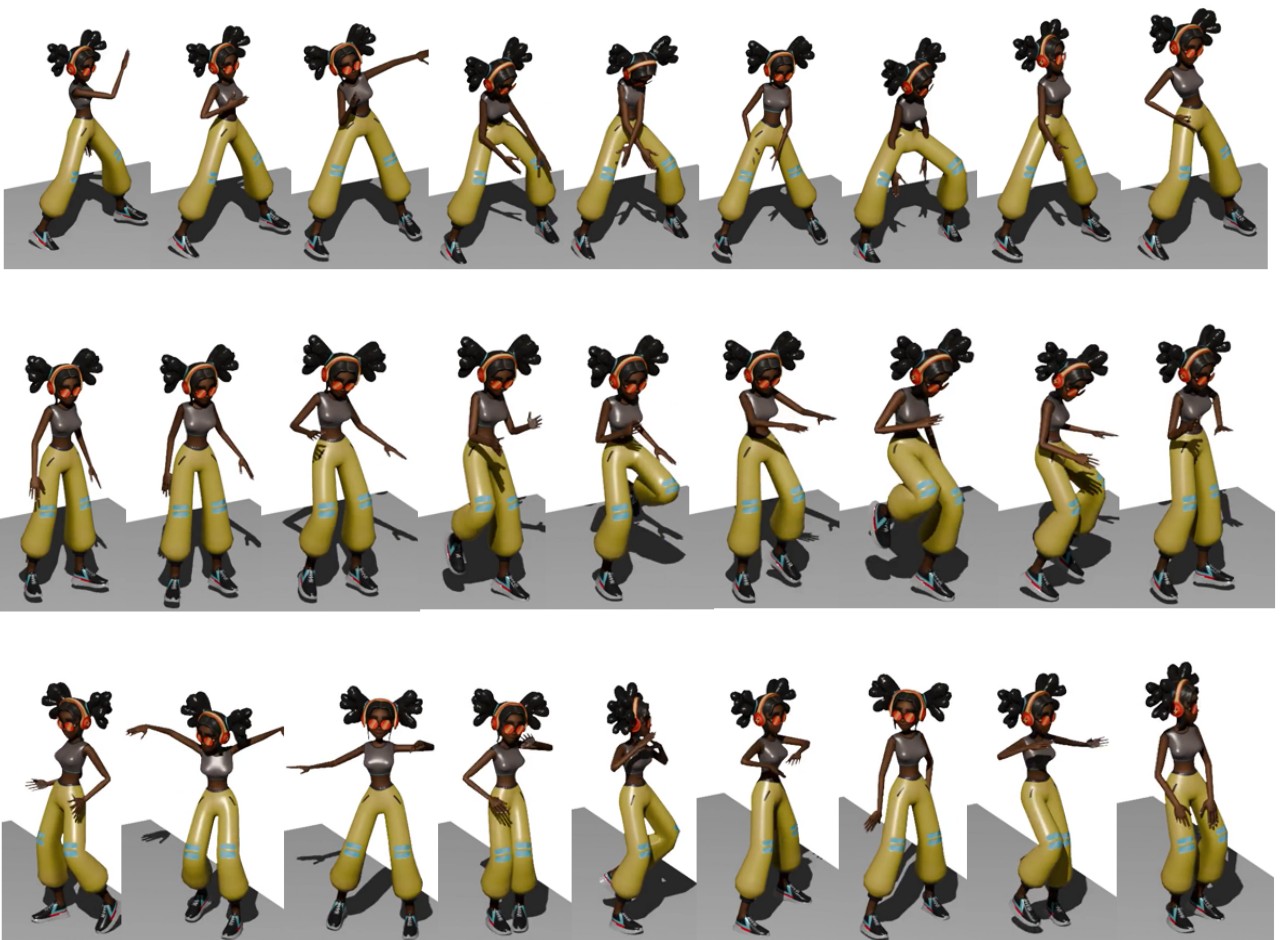

*Figure 6.* Qualitative comparison of audio-driven dance motion generated by different methods. From top to bottom: EDGE, Lodge, and Ours. EDGE produces aggressive motions but frequently exhibits self-interpenetration and mesh penetration artifacts. Lodge generates smooth yet overly conservative motions with limited amplitude and restricted body orientation changes. In contrast, Ours demonstrates larger and more diverse motion amplitudes, natural changes in body facing direction, and realistic full-body coordination while remaining physically feasible.

representations (Starke et al., 2022). For each pair, we compute the phase and amplitude as

$$\phi(t) = \frac{\text{atan2}(\sin(t), \cos(t))}{2\pi}, \qquad A(t) = \sqrt{\sin^2(t) + \cos^2(t)}, \tag{57}$$

which is a standard construction for extracting phase and magnitude from sinusoidal components (Starke et al., 2022). We select the dominant channel pair with the largest mean amplitude for visualization. By mapping phase and amplitude into Cartesian coordinates,

$$x(t) = A(t) \cdot \sin(2\pi\phi(t)), \qquad y(t) = A(t) \cdot \cos(2\pi\phi(t)), \tag{58}$$

we obtain a 2D phase manifold trajectory. To ensure a continuous and smooth trajectory, we visualize within a single window only, avoiding phase discontinuities and amplitude shifts caused by crossing window boundaries. The results show that PAE learns highly regular phase representations that form smooth spiral trajectories, and different audio clips yield separated spirals, indicating that PAE successfully disentangles periodic rhythmic patterns into a phase space.

**Vector-Quantized Periodic Autoencoder (VQPAE).** We use a 6-second window (96,000 samples) and extract phase from the encoder output $z_{\text{pae}}$ *before* vector quantization. The phase and amplitude are computed in the same way as in PAE, and we again visualize within a single window to avoid boundary-induced phase and amplitude biases. Since the VQPAE

encoder adopts a stride-1 design, the latent sequence has the same temporal length as the input; therefore, we apply a larger subsampling ratio (one point per 800 steps) to keep the plot legible. We further remove a small number of outliers (typically fewer than 5 points) using the interquartile range (IQR) rule to improve visualization clarity. The learned phase manifold of VQPAE exhibits a more compact and dense circular structure. Compared with the spiral trajectories of PAE, VQPAE features are more concentrated, implying a higher feature density within the same phase-space region. Such dense phase representations can provide stronger and more informative conditioning signals for downstream diffusion models (Chen et al., 2024a), which is beneficial for generating dance motions that better align with musical rhythms.

**Summary.** Overall, this visualization study highlights the distinctive advantages of periodic autoencoders (PAE/VQPAE) over raw waveforms and hand-crafted acoustic features. The learned phase representations have a clear geometric meaning and manifest as regular circular or spiral manifolds in 2D, in sharp contrast to the scattered distributions produced by the baselines. These results support that periodic autoencoders can effectively capture the intrinsic periodic structure of audio signals and provide structured audio representations for music-to-dance generation.

### C.2. Qualitative Music-to-Dance Comparisons

We qualitatively compare our method with representative diffusion-based baselines under the same audio conditions. As shown in Fig. 6, EDGE often produces visually aggressive motions, but suffers from noticeable self-interpenetration and mesh penetration artifacts, particularly during large upper-body swings and rapid limb movements. These artifacts indicate insufficient physical consistency when executed in a streaming rollout.

Lodge generates temporally smooth motions with strong continuity, but its movements tend to be overly conservative. The character exhibits limited motion amplitude and restricted changes in body orientation, resulting in reduced expressiveness and weaker alignment with dynamic musical cues.

In contrast, our method achieves a favorable balance between expressiveness and physical plausibility. The generated motions demonstrate larger and more diverse movement amplitudes, frequent and natural changes in body facing direction, and richer full-body coordination, while remaining visually realistic and physically feasible. This highlights the advantage of our streaming diffusion-forcing formulation in capturing diverse, music-responsive behaviors without sacrificing stability or executability.

### C.3. Interactive Avatar Platform

We build an online avatar platform to qualitatively evaluate real-time user-facing behavior under strict streaming constraints. The streaming generator produces continuous full-body motion conditioned on live audio, which is transmitted to a Unity-based visualization front-end and rendered without lookahead or post-hoc smoothing. As shown in Fig. 7, the avatar responds promptly to changes in musical rhythm while maintaining temporal coherence and natural pose transitions across long rollouts. This platform enables interactive inspection of responsiveness, stability, and visual quality in a deployment-faithful setting, bridging offline evaluation and real-time embodied execution.

### C.4. Physics-Based Humanoid Platform

We further deploy the streaming motion outputs on a physics-based humanoid platform to validate deployability beyond avatar visualization. The generated SMPL-format motions are retargeted online to the humanoid and converted into joint-space reference trajectories, which are executed by a real-time whole-body controller with low-level tracking. As shown in Fig. 8, the humanoid reproduces expressive upper-body movements and coordinated full-body poses while maintaining balance and temporal continuity. This demonstrates that our streaming generator produces physically consistent, low-latency motions that can be executed reliably in a closed-loop physics environment, supporting interactive and long-horizon operation.

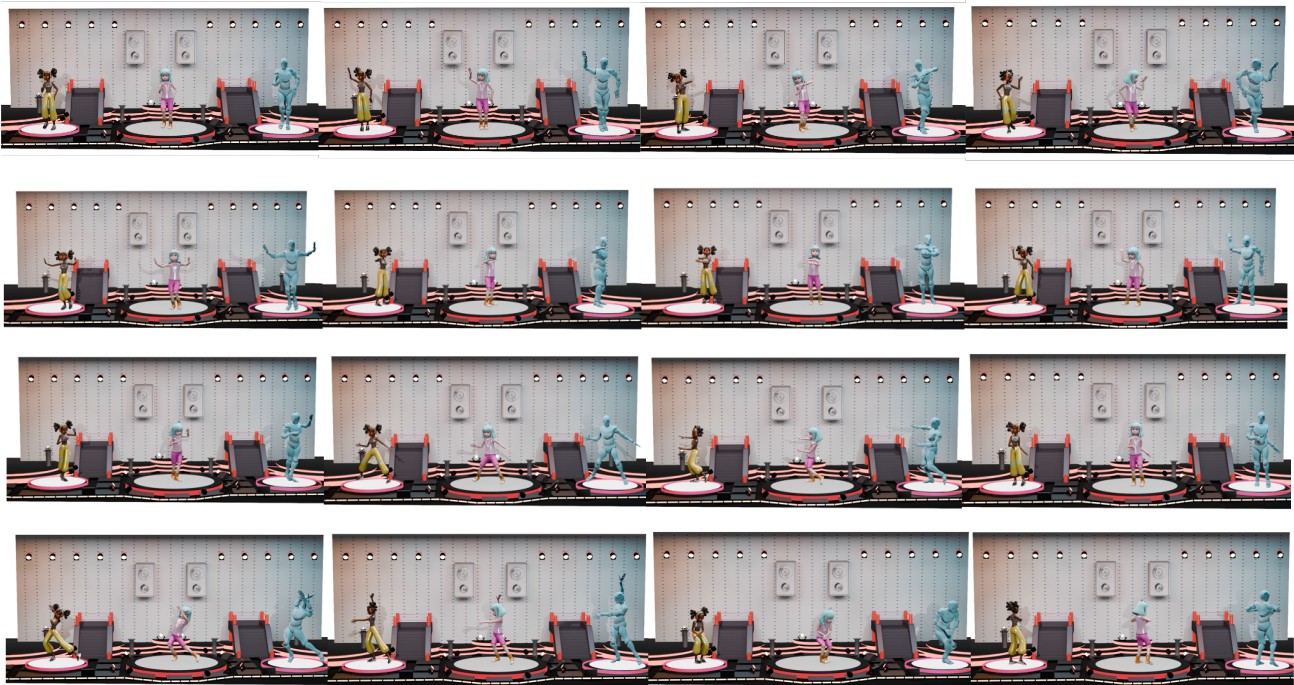

*Figure 7.* Real-time online avatar interaction driven by streaming audio-conditioned motion. Each row shows a continuous sequence of poses rendered in Unity, illustrating responsive upper-body gestures, coordinated full-body movement, and smooth transitions under strictly causal, low-latency control.

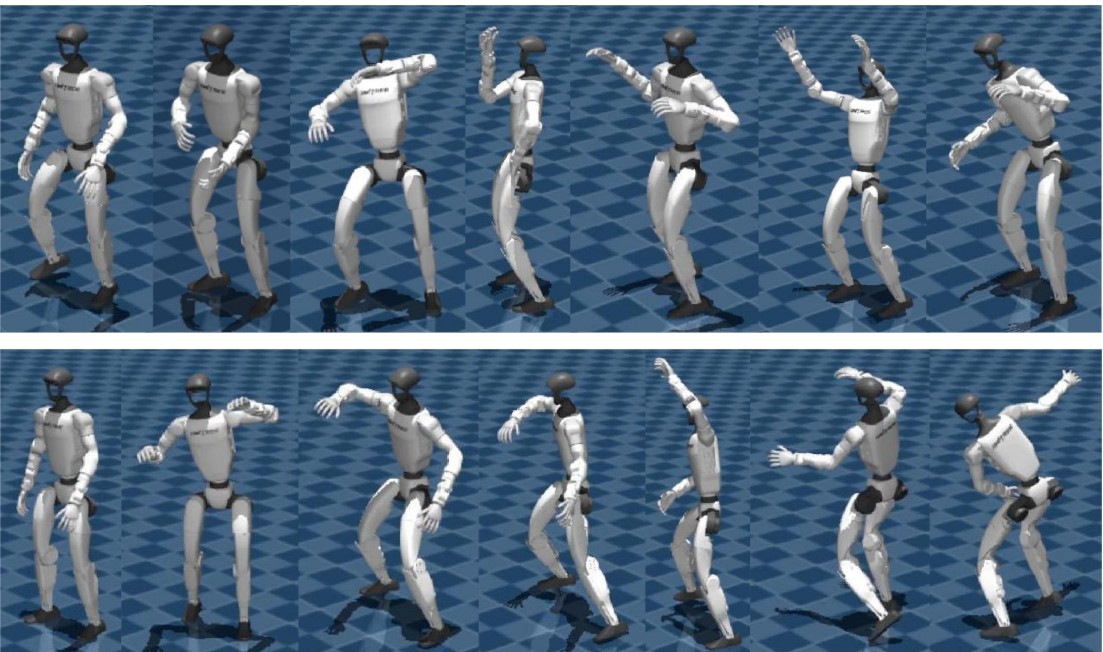

*Figure 8.* Physics-based humanoid execution driven by streaming audio-conditioned motion. Two representative motion sequences are visualized from multiple viewpoints. The humanoid tracks the generated whole-body reference in real time, exhibiting coordinated upper-body gestures, smooth transitions, and stable balance under continuous control.

---

**Algorithm 6** Streaming Motion Recovery for 272-D Representation

---

1: **Input:** Stream of frames $\{\mathbf{x}^{(t)}\}_{t\geq 0}$, where $\mathbf{x}^{(t)} \in \mathbb{R}^{272}$.

2: **Hyperparams:** Number of joints $J$ (e.g., 22), smoothing coefficient $\alpha \in [0,1]$, enable_smpl.

3: **Output (per frame):** World-space joints $\mathbf{P}^{(t)} \in \mathbb{R}^{J\times 3}$; optionally SMPL pose $\boldsymbol{\xi}^{(t)} \in \mathbb{R}^{3J}$.

4: **State:** Heading matrix $\mathbf{H} \in \mathbb{R}^{3\times 3}$, root translation $\mathbf{r} \in \mathbb{R}^3$, delayed delta $\mathbf{D} \in \mathbb{R}^{3\times 3}$, smoothed joints $\tilde{\mathbf{P}} \in \mathbb{R}^{J\times 3}$ (optional).

5: **Initialize:** $\mathbf{H} \leftarrow \mathbf{I}_3$, $\mathbf{r} \leftarrow \mathbf{0}$, $\mathbf{D} \leftarrow \mathbf{I}_3$, $\tilde{\mathbf{P}} \leftarrow \emptyset$.

6: **while** streaming **do**

7:     Receive current frame $\mathbf{x} \leftarrow \mathbf{x}^{(t)}$.

8:     **Parse 272-d layout:**

9:         $\mathbf{v}_{\text{nh}} \leftarrow (x_0, x_1) \in \mathbb{R}^2$                                                   $\triangleright$ root vel in XZ, no-heading

10:         $\mathbf{d}_{\text{6d}} \leftarrow \mathbf{x}_{2:7} \in \mathbb{R}^6$                                               $\triangleright$ heading delta rot6d

11:         $\mathbf{p}_{\text{nh}} \leftarrow \text{reshape}(\mathbf{x}_{8:8+3J-1}, (J, 3))$                         $\triangleright$ joint pos, no-heading

12:     *// 1) Convert heading delta*

13:     $\mathbf{D}_{\text{curr}} \leftarrow \text{Rot6DToMat}(\mathbf{d}_{\text{6d}}) \in \mathbb{R}^{3\times 3}$.

14:     *// 2) Integrate root translation using previous heading* $\mathbf{H}_{t-1}$

15:     $\mathbf{v} \leftarrow (v_{\text{nh},x}, 0, v_{\text{nh},z}) \in \mathbb{R}^3$.

16:     $\mathbf{v}_{\text{world}} \leftarrow \mathbf{H}^\top \mathbf{v}$                                        $\triangleright \mathbf{H}^{-1} = \mathbf{H}^\top$ for rotations

17:     $\mathbf{r} \leftarrow \mathbf{r} + \mathbf{v}_{\text{world}}$;   $\mathbf{r}_y \leftarrow 0$.

18:     *// 3) Delayed heading update (offline-compatible accumulation)*

19:     $\mathbf{H} \leftarrow \mathbf{D}\,\mathbf{H}$                                            $\triangleright$ apply previous frame's delta

20:     *// 4) Recover joints in world coordinates*

21:     $\mathbf{P}_{\text{root}} \leftarrow (\mathbf{H}^\top \mathbf{p}_{\text{nh}}^\top)^\top \in \mathbb{R}^{J\times 3}$                          $\triangleright$ undo heading

22:     $\mathbf{P} \leftarrow \mathbf{P}_{\text{root}}$.

23:     $\mathbf{P}_{:,x} \leftarrow \mathbf{P}_{:,x} + r_x$;   $\mathbf{P}_{:,z} \leftarrow \mathbf{P}_{:,z} + r_z$.

24:     *// 5) Optional EMA smoothing*

25:     **if** $\alpha < 1$ **then**

26:         **if** $\tilde{\mathbf{P}} = \emptyset$ **then**

27:             $\tilde{\mathbf{P}} \leftarrow \mathbf{P}$.

28:         **else**

29:             $\mathbf{P} \leftarrow \alpha \mathbf{P} + (1-\alpha)\tilde{\mathbf{P}}$.

30:             $\tilde{\mathbf{P}} \leftarrow \mathbf{P}$.

31:         **end if**

32:     **end if**

33:     *// 6) Store current delta for next frame*

34:     $\mathbf{D} \leftarrow \mathbf{D}_{\text{curr}}$.

35:     *// 7) Emit*

36:     **if** enable_smpl **= false then**

37:         Emit joints $\mathbf{P}$.

38:     **else**

39:         *// Optional SMPL: recover per-joint axis-angle*

40:         $\mathbf{R}_{\text{6d}} \leftarrow \text{reshape}(\mathbf{x}_{8+6J:8+12J-1}, (J, 6))$.

41:         $\mathbf{R} \leftarrow \text{Rot6DToMatBatch}(\mathbf{R}_{\text{6d}}) \in \mathbb{R}^{J\times 3\times 3}$.

42:         $\mathbf{R}_0 \leftarrow \mathbf{H}^\top \mathbf{R}_0$                                  $\triangleright$ apply heading to root rotation

43:         $\boldsymbol{\xi} \leftarrow \text{MatToAxisAngleBatch}(\mathbf{R}) \in \mathbb{R}^{3J}$.

44:         $\mathbf{t}_{\text{root}} \leftarrow (r_x, P_{0,y}, r_z)$.

45:         Emit $\{\text{joints} = \mathbf{P}, \text{pose\_aa} = \boldsymbol{\xi}, \text{root\_rot} = \boldsymbol{\xi}_{0:2}, \text{root\_trans} = \mathbf{t}_{\text{root}}\}$.

46:     **end if**

47: **end while**

48: **Helper:** $\text{Rot6DToMat}(\cdot)$ converts 6D rotation to $\mathbb{R}^{3\times 3}$ by Gram–Schmidt:

49:     $\mathbf{a}_1, \mathbf{a}_2 \leftarrow (d_{0:2}, d_{3:5})$; $\mathbf{b}_1 \leftarrow \text{norm}(\mathbf{a}_1)$; $\mathbf{b}_2 \leftarrow \text{norm}(\mathbf{a}_2 - (\mathbf{b}_1^\top \mathbf{a}_2)\mathbf{b}_1)$; $\mathbf{b}_3 \leftarrow \mathbf{b}_1 \times \mathbf{b}_2$; $\mathbf{R} \leftarrow [\mathbf{b}_1, \mathbf{b}_2, \mathbf{b}_3]$.

---

