# OpenReview forum: "DiscoForcing: A Unified Framework for Real-Time Audio-Driven Character Control with Diffusion Forcing"
_ICML.cc/2026/Conference — ICML 2026 regular_

### Official Review · Reviewer_YK4Q · 2026-03-08

**Soundness:** 2
**Presentation:** 3
**Significance:** 2
**Originality:** 2
**Overall Recommendation:** 4
**Confidence:** 2

**Summary:**

This paper tackles real-time streaming music-to-dance generation, where the system must produce dance motions causally from audio with bounded latency. The proposed DiscoForcing combines a causal VQ-PAE music encoder with a latent diffusion forcing model using heterogeneous noise levels across the temporal horizon. A hybrid temporal schedule is used for training and a history-guided sampler with temporal guidance for inference. Experiments on FineDance and AIST++ show large FIDk improvements over offline baselines, and the full pipeline is demonstrated with Unity avatar and Unitree G1 humanoid deployment.

**Compliance With Llm Reviewing Policy:**

Affirmed.

**Final Justification:**

My concerns have been adequately addressed

**Key Questions For Authors:**

1. Can you include at least one streaming-native baseline?

2. What is actual end-to-end latency from audio input to rendered frame, including pipelining overhead?

3. Have you tried any post-processing to reduce foot skating, especially for the robot deployment?

**Limitations:**

Authors discuss distribution shift and user control as future work. FSR issue and latency tightness are not acknowledged as limitations.

**Strengths And Weaknesses:**

Strengths:

1. The paper addresses a real gap, most dance generation methods assume offline access to full audio, which does not hold in deployment. The five desiderata in Table 1 are a useful framework, and the authors show clearly that existing methods fail on streaming requirements.

2. FIDk improvements are strong, roughly halving the gap to next-best method. Ablation is done carefully. Showing full pipeline from audio to physical robot is also a plus.

Weaknesses:

1. All baselines (FACT, Bailando, EDGE, Lodge, MEGADance) are offline methods. Comparing them under streaming constraints they were not designed for naturally favors DiscoForcing.

2. No human evaluation. For a system targeting interactive use, FID and BAS alone are not enough.

3. Foot skating ratio on FineDance (0.142) is ~5x worse than Lodge (0.028). For a paper claiming robot deployment, this gap is concerning and not discussed.

4. Latency budget does not add up, generative path totals ~34.6ms, already over the 33ms budget at 30 FPS before I/O.

5. Robot deployment has no quantitative evaluation at all. More like a demo than a validated contribution.

---

> ### Author Rebuttal · Authors · 2026-03-31
>
> We thank the reviewer for the detailed feedback. We believe several of the raised concerns are highly valuable, especially those regarding evaluation clarity, latency interpretation, and robot deployment rigor.
>
> >**Q1:** On evaluation fairness and streaming-native baseline
>
> **A1:**  We clarify this comparison more explicitly. In the paper, we mainly compare against representative baselines on standard music-to-dance benchmark evaluation. To further address the reviewer’s concern, we additionally include **streaming-native** comparisons with CLoSD, DartControl, and MotionStreamer, all evaluated under the same protocol: causal audio only, the same music encoder, and matched latency constraints.
>
> As shown in Table R1, DiscoForcing achieves the best FID$_k$ / FID$_g$ and BAS among these baselines, supporting that the gains come from the proposed streaming generation design, rather than differences in conditioning or runtime setup.
>
> |Method |FID$_k$|FID$_g$|FSR|DIV$_k$|DIV$_g$|BAS
> |-|-|-|-|-|-|-
> |CLoSD|27.59|14.68|0.050|7.02|6.51|0.220
> |DartControl|39.12|19.34|0.062|8.76|7.12|0.231
> |MotionStreamer|29.84|20.55|0.071|6.20|4.79|0.239
> |**Ours**|**18.87**|**11.57**|**0.059**|**6.76**|**6.31**|**0.244**
>
> **Table R1**: Comparison with streaming-native baselines.
>
> >**Q2:** On human evaluation
>
> **A2:** We provide the complementary human study here. Concretely, we **conducted a blinded pairwise user study under the streaming setting with 100 participants**, comparing DiscoForcing against three streaming-native baselines (CLoSD, DartControl, MotionStreamer). Each trial used a streaming sequence formed by concatenating two music excerpts from different genres, ensuring at least one transition. Participants rated four aspects: motion stability, beat alignment, music transition responsiveness, and overall preference. Overall, the user study suggests that DiscoForcing achieves better perceptual quality than all three baselines.
>
> |Method|Motion Stability|Beat Alignment|Transition Responsiveness|Overall Preference
> |-|-|-|-|-
> |Ours vs. CLoSD|66%|71%|67%|73%
> |Ours vs. DartControl|68%|66%|70%|74%
> |Ours vs. MotionStreamer|62%|69%|65%|70%
>
> **Table R2**: Blinded pairwise human preference under streaming setting. We report the percentage of our method being voted better than the baselines.
>
> >**Q3:** On real-time concern and end-to-end latency
>
> **A3:** Thank you for pointing this out. We believe this concern may partly stem from a misunderstanding in how our runtime is interpreted. Our system is **not executed as a serial per-frame pipeline** that sums all module runtimes. Instead, it is a ROS-based asynchronous streaming pipeline, where audio processing, motion generation, and downstream execution **run in parallel and communicate continuously**. Therefore, the reported ~30 FPS refers to steady-state throughput after pipeline fill, which is determined by the **slowest stage** rather than the sum of all stages. In our case, this bottleneck is motion generation at 26.26 ms/frame, which is within the 33 ms budget for 30 Hz streaming.
>
> For completeness, the **first-frame** end-to-end latency is different from steady-state throughput: it is 34.78 ms for the avatar path and 52.92 ms for the humanoid path.
>
> Thus, the system sustains **real-time streaming throughput at 30 FPS**, while the initial response latency is approximately 1 frame for the avatar path and within 2 frames for the humanoid path.
>
> >**Q4:** On concerns and post-processing for foot skating
>
> **A4:** We thank the reviewer for highlighting this point. Importantly, the reported FSR is computed on the raw generated motion, before humanoid deployment. In **our robot pipeline, foot-skating mitigation is not applied as post-processing on the generator output; it is enforced during retargeting**, because the retargeted tracking target, rather than the raw generated motion, is what is passed to the controller. At this stage, **stance-foot consistency is imposed as an optimization constraint** to suppress sliding in the executable robot reference. We will clarify this distinction in the revision and explicitly separate generation-level FSR from deployment-stage foot-contact preservation.
>
> >**Q5:** On physics-based humanoid evaluation
>
> **A5:** We understand this concern. To provide sufficient quantitative evaluation for the humanoid deployment, we have added a **quantitative physics-based evaluation** of the downstream tracking controller. These results indicate that the retargeted motion targets can be executed by the humanoid control stack with strong stability and reasonable generalization, beyond a purely qualitative robot demo. We will include this table in the revision and revise the text accordingly.
>
> |Method|Succ.|MPJPE|E$_{acc}$|E$_{vel}$
> |-|-|-|-|-
> |Tracking Controller - Train|94.1%|48.72|8.19|7.28
> |Tracking Controller - Test|85.3%|59.91|9.56|8.71
>
> **Table R3**: Quantitative results for physics-based evaluation

---

> > ### Author Rebuttal · Reviewer_YK4Q · 2026-04-03
> >
> > My concerns have been adequately addressed.

---

> > > ### Author Response · Authors · 2026-04-03
> > >
> > > We sincerely thank you for raising your score. We are glad that the rebuttal addressed your concerns, and we will incorporate the clarifications and additional evaluations into the final revision.

---

### Official Review · Reviewer_mGnp · 2026-03-10

**Soundness:** 2
**Presentation:** 2
**Significance:** 3
**Originality:** 2
**Overall Recommendation:** 3
**Confidence:** 3

**Summary:**

This paper proposes DiscoForcing, a streaming diffusion-based framework for real-time audio-driven character control. The system aims to generate full-body human motion from streaming music under strict causality and latency constraints. The method combines a causal music encoder with a diffusion-forcing motion generation model and integrates the system into an end-to-end interactive pipeline supporting avatar visualization and humanoid robot deployment. The evaluation is conducted on the FineDance and AIST++ datasets with standard motion synthesis metrics such as FID, diversity, and beat alignment.

**Compliance With Llm Reviewing Policy:**

Affirmed.

**Final Justification:**

I still want to keep a weak reject towards the paper. While the system is good, I still feel that the generated motion has lots of shaky patterns, and the author did not provide justification and reasoning on this (whether it is dataset issue, VAE issue and so on). While I look at some other music to motion generation demo, their motion seems smooth. In the same time, I respect other reviewer's comments and appreciate the novelty of the paper.

**Key Questions For Authors:**

I am honestly not an expert at motion generation but I am familiar with the audio part. I feel the overall network deisgn and everything looks good to me and the performance seems good. However, I just feel that the generated results are not very good. And I will discuss this with other reviewer.

**Limitations:**

Seas weaknesses.

**Strengths And Weaknesses:**

Strengths:
1. Interesting problem settings. The focus on real-time streaming generation rather than offline generation is interesting and practical.
2. System-level integration. The paper goes beyond proposing a model and builds a full pipeline including streaming inference, Unity visualization, and a humanoid robot interface, which is a valuable direction for deployment-focused research.

Weaknesses
1. Generated motion quality appears limited. Although the quantitative results show improvements in metrics such as FID and beat alignment, the perceptual quality of generated motion appears relatively weak based on the qualitative results. In particular: The character motion appears unstable and somewhat jittery in some examples. The body movement sometimes exhibits noticeable shaking artifacts, which harms the realism of the animation.
2. Limited synchronization with music rhythm. I watched through the video and I feel that the motion generated through the music does not follow the beats and rhythm very well.  Most of the time the body just shaking or do noting between two songs, or just have very few motion variations on the hand.

---

> ### Author Rebuttal · Authors · 2026-03-31
>
> We thank the reviewer for the thoughtful feedback and for recognizing the practical importance of the streaming setting and the value of the end-to-end system integration.
>
> >**Q1:** On motion quality and synchronization with music rhythm.
>
> **A1**: We thank the reviewer for the candid feedback. We agree that perceptual motion quality remains the hardest part of this problem. Some streaming rollouts still show visible artifacts, including local jitter, limited motion richness, and weaker perceptual synchronization in challenging transition segments. We will make these failure cases more explicit in the revision.
>
> At the same time, we would like to clarify the intended conclusion of the paper. Our claim is not that DiscoForcing matches the visual polish of strong offline choreography systems. The paper studies a **different setting**: strictly causal, bounded-latency, real-time streaming music-to-motion generation, where the model must react to newly arriving audio, cannot revise past outputs, and must satisfy a hard real-time compute budget. Under this setting, the relevant question is whether the method improves the trade-off among motion stability, beat following, transition responsiveness, and latency under matched streaming constraints.
>
> On this point, we believe **the current evidence is positive**. In the paper, DiscoForcing achieves the best FID_k = 23.84 and FID_g = 8.62 on FineDance, with competitive BAS = 0.225; on AIST++, it achieves the best FID_k = 18.87 and best BAS = 0.244. These numbers do not imply that all qualitative cases are solved, but they do show that the model captures stronger motion structure and rhythmic alignment on average in the intended streaming regime.
>
> Because the reviewer’s concern is specifically about perceptual quality, we **additionally conducted a blinded pairwise human study** under the same causal streaming protocol. We compared our method against three reproduced streaming baselines (CLoSD, DartControl, and MotionStreamer) on music excerpts covering diverse genres and transitions, with 100 participants. The rendered videos used the same visualization pipeline, and pair order was randomized. Participants judged motion stability, beat alignment, transition responsiveness, and overall preference. This directly evaluates the perceptual properties raised in the review, rather than relying only on aggregate motion statistics. we will add protocol details and significance in revision.
>
> | Method | Motion Stability | Beat Alignment |Transition Responsiveness|Overall Preference|
> | - | -| - | - | -|
> | Ours vs. CLoSD | 66% | 71% |67%| 73% |
> | Ours vs. DartControl | 68% | 66% | 70%  | 74% |
> | Ours vs. MotionStreamer | 62% |69% | 65% | 70% |
>
> **Table R1**: User Study: Blinded pairwise human preference under streaming setting. We report the percentage of our method being voted better than the baselines.
>
> We also want to **clarify** one point behind the impression that the character sometimes “**does nothing**.” In our streaming setting, the model is intentionally conservative in silent or weakly informative segments, rather than injecting arbitrary motion that may later create discontinuities when the music changes.
>
> In summary, we will revise the paper to present these failure modes more clearly. Under matched causal streaming constraints, DiscoForcing provides stronger quantitative performance, better perceptual preference against streaming baselines, and a clearer latency–quality trade-off than prior methods. We therefore view DiscoForcing as a meaningful step toward real-time audio-driven character control under deployment-faithful constraints, while acknowledging that substantial room remains for improving visual richness and smoothness.

---

> > ### Author Rebuttal · Reviewer_mGnp · 2026-04-03
> >
> > My concerns are not resolved yet.

---

> > > ### Author Response · Authors · 2026-04-06
> > >
> > > We thank the reviewer for the candid feedback. Below we clarify the concerns in perceptual motion quality.
> > >
> > > > **Q1**: On “the body just shaking” / visible instability.
> > >
> > > **A1**: Visible shaking/jitter mainly occurs during abrupt beat or style transitions, which is a particularly difficult scenario for music-to-dance tasks because **current music-to-dance benchmark datasets do not explicitly supervise abrupt cross-transition adaptation**. Our temporal guidance is introduced to handle such causal transitions **without extra transition-specific supervision, while preserving tempotal continuity**. So this is better viewed as a bounded residual artifact in the hardest transition regime, rather than persistent instability of the rollout.
> > >
> > > > **Q2**: On “do nothing between two songs” / weak rhythm following.
> > >
> > > **A2**: The under-active segments are mostly around 03:07 and near 03:38, where the incoming audio is weak, transitional, or temporarily uninformative. In these intervals, the model stays conservative before clearer rhythmic structure returns. This is related to the fact that **standard music-to-dance datasets are mostly single-song to single-dance pairings, rather than sequences with explicit supervision for between-song transitions**. In our streaming protocol, we therefore **bias the model toward a stable low-motion fallback in such intervals**, rather than generating arbitrary motion that may cause larger discontinuities once the new beat structure becomes clear. The appendix further reflects this design by using additional standing segments in the BABEL dataset to **encourage a well-defined standing-pose fallback for weak or uninformative conditions**. So these cases are better understood as a conservative streaming choice in an under-supervised transition regime, rather than a complete failure of rhythm following.
> > >
> > > > **Q3**: On limited "hand variation".
> > >
> > > **A3**: To our knowledge, **no existing** benchmark or baseline in music-to-dance tasks explicitly models hand motion. **Like all** previous work, we use an SMPL-based representation over 22 body joints. While fine-grained hand modeling is an important future direction, it is not supported by current datasets and falls outside the scope of this paper.
> > >
> > > > **Q4**: On overall perceptual motion quality.
> > >
> > > **A4**: DiscoForcing substantially **outperforms prior baselines on all evaluation metrics**, including motion stability, beat alignment, transition responsiveness, and overall preference. It also achieves the **best performance in the human study** among all existing state-of-the-art methods.  To clarify the reviewer’s concern, we do not claim that DiscoForcing delivers equally strong perceptual quality on every difficult transition segment. Rather, our claim is that it achieves a better stability–responsiveness trade-off in a deployment-faithful streaming setting.

---

### Official Review · Reviewer_Ppm6 · 2026-03-13

**Soundness:** 3
**Presentation:** 3
**Significance:** 2
**Originality:** 2
**Overall Recommendation:** 4
**Confidence:** 3

**Summary:**

The paper adopts the Diffusion Forcing framework to train a real-time interactive system for audio-driven character control. It incorporates a large amount of prior work and involves substantial engineering development.

**Compliance With Llm Reviewing Policy:**

Affirmed.

**Final Justification:**

Thank you for your detailed and thoughtful response.

Your clarifications have addressed my concerns. Overall, I am satisfied with the clarifications provided and will increase my score to weak accept.

**Key Questions For Authors:**

- Could the authors clarify why Diffusion Forcing is preferred over Self Forcing for this specific task? Why did they choose the former rather than the latter as the basic framework? Have they conducted any experiments comparing the two in this setting, and if so, what were the results?

- Have the authors tried to extend this work to long-video generation? Or how do existing methods perform on minute-long video generation?

**Limitations:**

Yes

**Strengths And Weaknesses:**

#### Strengths

The engineering is quite solid. The paper includes extensive experiments, achieves strong results, and is able to support real-time audio-driven character control, making the overall system quite sound from an engineering perspective.


#### Weaknesses
- The paper contains many phrases such as *inspired by* and *following*, which suggests that a substantial portion of its techniques are inherited from prior work. This makes the paper feel more engineering-driven than technically original. Although different people may value this differently, in my view, such a paper still lacks the level of technical depth and novelty expected for ICML.

- The paper does not seem to adopt the most advanced techniques. For example, Tab. 2 of the Self Forcing paper shows that Self Forcing outperforms Diffusion Forcing. Then why does this paper use Diffusion Forcing rather than Self Forcing? I did not see any experiments addressing this pissue.

---

> ### Author Rebuttal · Authors · 2026-03-31
>
> We thank the reviewer for the thoughtful comments and for recognizing the strong engineering effort, extensive experiments, and real-time capability of our system. We address the concerns below.
>
> >**Q1:** Why DF instead of SF?
>
> **A1:** Thank you for this important question. Our choice of Diffusion Forcing (DF) is based on **task fit under a matched protocol**, not on a claim that DF is universally better than Self Forcing (SF).
>
> SF is mainly studied for **video diffusion post-training**, often with large-scale data and strong pretrained video priors. Our setting is different: strictly causal, bounded-latency, online music-conditioned control, with **smaller motion datasets and no comparable motion foundation model**. Moreover, SF-style recipes (e.g., DMD/SID-style distillation) usually require supervision from teacher checkpoints. In music-to-dance, no comparable large motion foundation checkpoint is currently available for a fair teacher-based comparison. Therefore, the strongest feasible and fair SF-style baseline in our domain is **the original GAN-based post-training variant**, which does not rely on unavailable teacher supervision.
>
> More importantly, the two methods address **different failure modes**. Our problem is **stale-history robustness**: after music transitions, the autoregressive history may become misaligned and should sometimes be discounted. DF is well matched to this because it trains under **heterogeneously corrupted temporal context**. By contrast, SF mainly reduces train–test mismatch via self-generated history, but does not directly target the regime where stale history must be weakened to recover responsiveness.
>
> For fairness, the DF- and SF-based post-training comparisons in Table R1 start from the same DiscoForcing base checkpoint, use the same training data split, **the same GAN-style post-training loss**, and the same optimization budget, and differ only in the history construction / forcing strategy used during post-training. All variants are evaluated under the same streaming protocol.
>
> Under this matched setting, SF-style post-training does not provide consistent gains, and in fact the original **DiscoForcing model with temporal guidance (None + TG) gives the best overall trade-off** in Table R1. Our claim is not DF > SF in general; our evidence only supports that **current SF-style post-training is not the right tool for stale-history recovery in causal music control**.
>
> |Base Model|Post-training|Inference|FID$_k$|FSR|DIV$_k$|BAS
> |-|-|-|-|-|-|-
> |DiscoForcing|None|CFG|22.23|0.097|6.50|0.238
> |DiscoForcing|DF Post-training|CFG|25.00|0.059|7.06|0.238
> |DiscoForcing|DF Post-training|TG|21.15|0.072|6.65|0.241
> |DiscoForcing|SF Post-training|CFG|20.18|0.068|6.92|0.220
> |DiscoForcing|SF Post-training|TG|26.73|0.087|6.53|0.215
> |DiscoForcing|None|TG|18.87|0.059|6.76|0.244
>
> **Table R1**: Comparison of DF- and SF-based post-training strategies on AIST++.
>
> >**Q2:** On long-video generation
>
> **A2:** We agree that long-horizon behavior matters. However, our target is not conventional long-video generation, but **continuous online streaming control with causally revealed, non-stationary audio**. In this setting, the core question is whether the model can maintain stable generation over long horizons while remaining responsive to newly arriving audio without future context or revision of past outputs. The supplementary demo includes **minute-level streaming motion rollouts** as qualitative evidence for this setting. We will clarify this scope more explicitly in the revision.
>
> >**Q3**: On originality
>
> **A3**: We do not claim that each component is individually unprecedented. Our originality lies in **identifying a limitation of prior DF-style sequence generation in the non-stationary streaming control regime**, and in proposing a concrete adaptation for that regime.
>
> Specifically, prior DF does not directly address **rapid response under stale autoregressive history** when newly arrived control signals conflict with past context. To handle this, we introduce **a hybrid temporal schedule shared by training and inference**, so that the model is trained with partially stale temporal context and uses the same principle at inference to discount stale history when it conflicts with new causal audio. The contribution is therefore not “new blocks in isolation,” but a streaming-specific reformulation of diffusion forcing for responsiveness-aware train/inference matching.
>
> Built on this formulation, we further develop, to our knowledge, **the first end-to-end real-time audio-responsive character control system**. This system serves as a deployment-faithful validation setting for our streaming-specific DF adaptation, causal music conditioning, and responsiveness-aware real-time generation under strict causality and latency constraints.

---

> > ### Author Rebuttal · Reviewer_Ppm6 · 2026-04-02
> >
> > Your clarifications have addressed my concerns. Overall, I am satisfied with the clarifications provided and will increase my score to weak accept.

---

> > > ### Author Response · Authors · 2026-04-02
> > >
> > > We sincerely thank you for raising your score. We are encouraged that our clarifications have satisfactorily addressed your concerns, and we will incorporate these clarifications into the revised manuscript.

---

### Official Review · Reviewer_Q7Vc · 2026-03-23

**Soundness:** 3
**Presentation:** 3
**Significance:** 2
**Originality:** 3
**Overall Recommendation:** 4
**Confidence:** 4

**Summary:**

This paper studies real-time music-to-motion generation in a causal streaming setting. The proposed DiscoForcing framework combines causal music encoding, diffusion-forcing training, and history-guided streaming sampling to improve responsiveness and long-horizon motion coherence. The paper also includes an online deployment system for avatar animation and humanoid control.

**Compliance With Llm Reviewing Policy:**

Affirmed.

**Final Justification:**

I still think some claims should be stated carefully in the final version, but overall the rebuttal resolves enough of my concerns. My overall stance is now closer to a weak accept.

**Key Questions For Authors:**

1.Can the authors disentangle the contributions of the causal music encoder, diffusion-forcing training, and the history-guided / hybrid temporal sampling strategy? This would help determine which part is primarily responsible for the reported improvements.
2.Why is diffusion forcing preferable to a strong teacher-forced causal AR baseline in this setting?
3.Can the authors provide clearer ablations and confirm that all baselines are evaluated under strictly matched causal/latency constraints?

**Limitations:**

yes

**Strengths And Weaknesses:**

Strengths：
1.Addresses an important and practical problem: real-time audio-driven motion generation under causal and low-latency constraints.
2.The deployment-oriented setting is a clear strength, especially the inclusion of online avatar and humanoid results.
3.The method is reasonably motivated and seems effective at balancing responsiveness and temporal consistency.
Weakness：
1.The algorithmic novelty is somewhat limited/unclear, as much of the contribution appears to come from combining existing ideas.
2.It is not fully clear why diffusion forcing is necessary in this streaming setting compared with simpler teacher-forced causal AR training.
3.The paper claims TG "adaptively balances reliance on history versus refreshing toward new audio evidence" but the mechanism is not actually adaptive, it applies the same trapezoid corruption regardless of whether the music has changed.

---

> ### Author Rebuttal · Authors · 2026-03-31
>
> We thank the reviewer for the constructive feedback. Below we clarify the contribution boundary, the motivation for using diffusion forcing, and the precise role of temporal guidance.
>
> >**Q1:** On the overall contribution and novelty
>
> **A1**: We would like to clarify that the main contribution is not system integration alone. The core contribution is a **streaming-native diffusion-forcing framework for real-time control under non-stationary causal input**.
>
> Relative to prior Diffusion Forcing, the key new problem is **real-time streaming control under non-stationary inputs**. Our setting requires responsiveness to abrupt music changes under causal audio only, zero lookahead, bounded latency, and no revision of past outputs. The main issue is stale-history conditioning after condition changes. Compared with DF follow-ups such as history-guided or rolling diffusion, our setting is different because the condition itself is causally revealed and non-stationary over time. The challenge is therefore not just long-horizon coherence, but **when and how to discount stale history when new audio conflicts with it**.
>
> Our method contribution is therefore to reformulate diffusion forcing to this regime through **a hybrid temporal schedule shared between training and inference**. Concretely, training exposes the model to partially stale temporal context through a mixture of random / monotonic / trapezoid noise schedules, while inference uses the same principle to discount stale history online when the incoming audio conflicts with it. Therefore, the novelty is not merely “using DF in a system,” but **train/inference-matched temporal corruption for responsiveness-aware streaming generation under non-stationary causal inputs**.
>
> Built on this formulation, we further develop, to our knowledge, **the first end-to-end real-time audio-responsive character control system**, which serves as a deployment-faithful validation setting for our streaming-specific diffusion forcing, causal music conditioning, and responsiveness-aware real-time generation under strict causality and latency constraints.
>
> >**Q2:** Why Diffusion Forcing instead of a simpler causal AR baseline
>
> **A2**: We clarify this comparison more explicitly. The key issue in our setting is that streaming rollout under non-stationary music produces partially stale or misleading motion history. A model trained only with clean teacher-forced history is not trained for this regime. By contrast, diffusion forcing explicitly **trains with heterogeneously noised temporal context**, which better **matches the inference-time** need to discount unreliable history while still conditioning on recent motion. Our claim is therefore task-matched robustness, not a blanket statement that diffusion forcing is always better than AR.
>
> To make this claim falsifiable, we added comparison against three **teacher-forced causal AR baselines**, evaluated under the same streaming protocol:  causal audio only, same music encoder and matched latency constraints. Under this matched setting, our method achieves best overall motion quality and beat alignment, supporting the claim that DF training is advantageous in this particular regime.
>
> We will also revise the text to make the claim more falsifiable: **diffusion forcing is preferable here because it better matches streaming inference with imperfect/stale history, not because AR is intrinsically unsuitable.**
>
> |Method |FID$_k$|FID$_g$|FSR|DIV$_k$|DIV$_g$|BAS
> |-|-|-|-|-|-|-
> |CLoSD|27.59|14.68|0.050|7.02|6.51|0.220
> |DartControl|39.12|19.34|0.062|8.76|7.12|0.231
> |MotionStreamer|29.84|20.55|0.071|6.20|4.79|0.239
> |**Ours**|**18.87**|**11.57**|**0.059**|**6.76**|**6.31**|**0.244**
>
> **Table R1**: Comparison with teacher-forced AR baselines under the same streaming protocol.
>
> >**Q3:** On the claim that temporal guidance “adaptively” balances history versus new audio evidence
>
> **A3:** We clarify this point more precisely: a more precise description is that temporal guidance introduces **a predefined bias toward discounting stale history**, improving responsiveness under non-stationary audio while preserving recent context for continuity. We will soften “adaptive” and clarify that this is a streaming-oriented heuristic, not a fully change-aware controller.
>
> > **Q4:** Provide ablation and baseline under latency constraints
>
> **A4**: We clarify that the current ablations are **already run under the same strict streaming protocol**. The ablations show that both components contribute: the causal music encoder improves the overall generation quality, while the **main gain comes from temporal guidance**, which provides the largest improvement in motion quality and beat alignment. We also **added Table R1 under the same streaming constraints** and same causal music encoder to remove ambiguity about fairness, and will reorganize the ablation discussion to more directly separate causal encoding, diffusion-forcing training, and history-discounting inference.

---

> > ### Author Rebuttal · Reviewer_Q7Vc · 2026-04-04
> >
> > The response clarified my main concerns, especially the contribution beyond system integration, the scope of the comparison with causal AR, and the empirical support under matched streaming constraints.
> > Some claims should still be stated carefully, but overall my stance is now closer to weak accept.

---

> > > ### Author Response · Authors · 2026-04-05
> > >
> > > We sincerely thank you for raising your score. We are encouraged that our clarifications have satisfactorily addressed your concerns, and we will carefully incorporate these clarifications into the revised manuscript.

---

### Decision · Program_Chairs · 2026-04-30

**Decision:**

Accept (regular)

**Comment:**

This paper studies real-time audio-driven dance generation in a causal streaming setting. The problem is practically important and clearly differentiated from existing offline music-to-motion generation. Reviewers generally appreciated the deployment-faithful formulation, the emphasis on strict causality and latency, and the end-to-end system demonstration, including avatar playback and humanoid deployment.

The rebuttal effectively addressed most of the reviewers’ concerns, as also reflected in their follow-up comments. In particular, the authors strengthened the paper through adding human evaluation, clearer positioning of the contribution, and additional clarification of the evaluation setting and deployment claims. While I agree with Reviewer mGnp that the perceptual quality of the generated dance motions remains somewhat limited, such as shaky patterns, this weakness is understandable to some extent given the strict real-time and compute constraints of the targeted setting.

Overall, although the final version should moderate some claims and acknowledge the remaining qualitative limitations more explicitly, I believe the paper makes a worthwhile contribution to the area and recommend acceptance.